# Isolation, culture, and characterisation of bovine ovarian fetal fibroblasts and gonadal ridge epithelial-like cells and comparison to their adult counterparts

Menghe Liu[1], Katja Hummitzsch[1], Nicole A. Bastian[1], Monica D. Hartanti[1,2], Qianhui Wan[1], Helen F. Irving-Rodgers[1,3], Richard A. Anderson[4], Raymond J. Rodgers[1] *

1 School of Biomedicine, Robinson Research Institute, The University of Adelaide, Adelaide, SA, Australia, 2 Faculty of Medicine, Universitas Trisakti, Jakarta, Indonesia, 3 School of Medical Science, Griffith University, Gold Coast Campus, QLD, Australia, 4 MRC Centre for Reproductive Health, Queen's Medical Research Institute, University of Edinburgh, Edinburgh, United Kingdom

* ray.rodgers@adelaide.edu.au

## Abstract

During ovarian development, gonadal ridge epithelial-like (GREL) cells arise from the epithelial cells of the ventral surface of the mesonephros. They ultimately develop into follicular granulosa cells or into ovarian surface epithelial cells. Stromal fibroblasts arise from the mesonephros and penetrate the ovary. We developed methods for isolating and culturing fetal ovarian GREL cells and ovarian fibroblasts by expansion of colonies without passage. In culture, these two cell types were morphologically different. We examined the expression profile of 34 genes by qRT-PCR, of which 24 genes had previously been studied in whole fetal ovaries. Expression of nine of the 10 newly-examined genes in fetal ovaries correlated with gestational age (*MUC1*, *PKP2*, *CCNE1* and *CCNE2* negatively; *STAR*, *COL4A1*, *GJA1*, *LAMB2* and *HSD17B1* positively). Comparison between GREL cells and fetal fibroblasts revealed higher expression of *KRT19*, *PKP2*, *OCLN*, *MUC1*, *ESR1* and *LGR5* and lower expression of *GJA1*, *FOXL2*, *NR2F2*, *FBN1*, *COL1A1*, *NR5A1*, *CCND2*, *CCNE1* and *ALDH1A1*. Expression of *CCND2*, *CCNE1*, *CCNE2*, *ESR2* and *TGFBR1* was higher in the fetal fibroblasts than in adult fibroblasts; *FBN1* was lower. Expression of *OCLN*, *MUC1*, *LAMB2*, *NR5A1*, *ESR1*, *ESR2*, and *TGFBR3* was lower in GREL cells than ovarian surface epithelial cells. Expression of *KRT19*, *DSG2*, *PKP2*, *OCLN*, *MUC1*, *FBN1*, *COL1A1*, *COL3A1*, *STAR* and *TGFBR2* was higher and *GJA1*, *CTNNB1*, *LAMB2*, *NR5A1*, *CYP11A1*, *HSD3B1*, *CYP19A1*, *HSD17B1*, *FOXL2*, *ESR1*, *ESR2*, *TGFBR3* and *CCND2* was lower in GREL cells compared to granulosa cells. TGFβ1 altered the expression of *COL1A1*, *COL3A1* and *FBN1* in fetal fibroblasts and epidermal growth factor altered the expression of *FBN1* and *COL1A1*. In summary, the two major somatic cell types of the developing ovary have distinct gene expression profiles. They, especially GREL cells, also differ from the cells they ultimately differentiate in to. The regulation of cell fate determination, particularly of the bi-potential GREL cells, remains to be elucidated.

**Data Availability Statement:** All the data that pertain to the results are presented in the text or supplementary figures and tables.

**Funding:** Funding was received from Adelaide University China Fee Scholarships (ML and QW; https://www.csc.edu.cn/chuguo), Adelaide Graduate Research Scholarship (NAB; http://www.adelaide.edu.au/), Australia Awards Scholarship (MDH; https://www.dfat.gov.au/people-to-people/australia-awards/australia-awards-scholarships), Robinson Research Institute Career Development Fellowship (KH; https://www.adelaide.edu.au/robinson-research-institute/) and Building On Ideas Grant (KH; https://www.adelaide.edu.au/robinson-research-institute/), National Health and Medical Research Council of Australia Centre for Research Excellence in the Evaluation, Management and Health Care Needs of Polycystic Ovary Syndrome (RJR; GTN1078444; https://www.nhmrc.gov.au/) and The Centre for Research Excellence on Women's Health in Reproductive life (RJR and KH; GTN1171592; https://www.nhmrc.gov.au/) and the UK Medical Research Council (RAA; grant No. G1100357; https://mrc.ukri.org/). The funders did not play any role in the study design, data collection and analysis, decision to publish, or preparation of the manuscript.

**Competing interests:** RAA reports consultancy work for Ferring, Merck, IBSA, Roche Diagnostics, NeRRe Therapeutics and Sojournix Inc. The other authors of this manuscript have nothing to declare and no conflict of interest that could be perceived as prejudicing the impartiality of the research reported. This does not alter our adherence to PLOS ONE policies on sharing data and materials.

## Introduction

Understanding the development of the mammalian ovary is important for identifying the origins of different ovarian cell types and ovarian diseases. Despite much research in this area, the knowledge of ovarian development is still somewhat disjointed, particularly across species. It is commonly accepted that the ovary arises from the mesonephros [1, 2], but the origins of somatic ovarian cells is not clear. For instance, granulosa cells that are epithelial cells that surround each oocyte in follicles have been proposed to arise from the rete ovarii [3–5], the ovarian surface epithelium [6, 7] or a recently identified cell type, the gonadal ridge epithelial-like (GREL) cell [8]. Based upon morphological observations GREL cells are not only a precursor of granulosa cells but also of ovarian surface epithelial cells [8, 9]. Penetration of stromal fibroblasts from the mesonephros into the primordial ovary has consistently been observed in a number of species [6, 8, 10, 11].

In the ovary, granulosa cells contribute supportive functions to the oocyte via gap junctions [12–14] and they communicate with the thecal cells [15, 16] that surround them. They play an important role in follicle selection and follicle development [17, 18]. Furthermore, enzymes which are essential to convert androgens into oestrogens [19, 20] are highly expressed in granulosa cells late in follicular development. On the other hand, the ovarian surface epithelium is a single layer of epithelial cells covering the surface of the ovary. It is involved in the recovery of the ovarian surface following the rupture of follicles at ovulation by cell proliferation. Lgr5-positive stem cells have been identified in the ovarian surface epithelium where they function as stem/progenitor cells for the recovery of the ovarian surface after ovulation [21, 22]. The differences between the two cell types which GREL cells differentiate into could be the result of regulation by oocyte-secreted factors such as growth differentiation factor 9 (GDF9) and bone morphogenetic protein 15 (BMP15) which play essential roles in regulating the proliferation and differentiation of granulosa cells [23]. The ovarian surface epithelium, on the other hand, is separated from oocytes by stroma and basal lamina and is likely not exposed to oocyte-secreted factors.

Stroma in the ovary provides structural support to this organ, but it also plays active roles. During the development of the ovary, stroma from the mesonephros penetrates into the primordial fetal ovary, and branches as it does so. This branching thus partitioning the GREL cells and oogonia thereby creates the ovigerous cords. The stroma further partitions the ovigerous cords into smaller groups of cells which become primordial follicles containing an oocyte, derived from oogonia, and granulosa cells, derived from GREL cells. The stroma penetrates to just below the surface of the ovary and spreads laterally and the GREL cells on the surface differentiate into ovarian surface epithelium [8, 24]. The stroma continues to play significant roles in adult ovaries. *In vitro* culture of follicles demonstrates that the strength and rigidity of an artificial matrix can influence the growth of follicles [25], which suggests that *in vivo* the strength and rigidity of stroma surrounding follicles can influence follicle development. The fibroblast is the major cell type in stroma, and hormones and growth factors can regulate the proliferation and extracellular matrix (ECM) production of fibroblasts [26].

To conduct the first examination of GREL cells and to compare them to fetal fibroblasts, we examined the expression of a range of genes related to epithelial cells, stromal cells, steroidogenesis, signalling receptors, cell cycle and germ/stem cells. We also examined the expression of these genes in the adult cells that the GREL cells and fetal fibroblast differentiate in to. We also examined potential regulators of expression of stromal-related genes in fetal fibroblast.

## Materials and methods

### Collection of bovine fetal and adult ovaries

Fetuses from pregnant *Bos taurus* cows and ovaries from non-pregnant *Bos taurus* cows were collected at local abattoirs (Thomas Foods International, Murray Bridge, SA, Australia and Strath Meats, Strathalbyn, SA, Australia) and transported on ice in Hank's balanced-salt solution (HBSS) with calcium and magnesium (HBSS$^{+/+}$) to the laboratory. The gestational ages of the fetuses were estimated from the crown-rump length (CRL) [27]. Fetuses with a CRL < 8 cm underwent a sex determination as previously reported [8]. Samples of whole fetal ovaries throughout gestation (n = 27) were from our previous study [28] and collected from The Midfield Group, Warrnambool, VIC, Australia. The bovine ovaries used in this study were scavenged from animals that were being processed for human consumption and were not owned by the authors or their institutions. As such the University of Adelaide's Animal Ethics Committee only requires notification of this.

### Isolation of fetal ovarian GREL cells and fibroblasts

Ovaries were dissected from the fetus and rinsed once with 70% ethanol and twice with HBSS$^{+/+}$. Then the fetal ovaries were fully minced and digested in 10 ml HBSS$^{+/+}$ containing 1 mg/ml collagenase type I (GIBCO, Carlsbad, CA, USA) and 50 μg/ml DNase I (Sigma-Aldrich, New South Wales, Australia) at 37˚C and shaking at 100 rpm. Minced ovaries from fetuses (CRL ≤ 20 cm) were digested for 30 min and ovaries from fetuses (CRL > 20 cm) were digested for 45 min. The cell suspensions were then passed through cell strainers and centrifuged at 1200 rpm for 5 min and the supernatants were discarded. Subsequently, the cell pellets were washed with 5 ml HBSS$^{+/+}$ and digested with 1 ml red blood cell lysis solution (Sigma-Aldrich, New South Wales, Australia; Cat # R7757) by gently pipetting up and down for 45 sec. Thereafter, cell pellets were resuspended in 1 ml DMEM/F12 medium (Sigma-Aldrich, New South Wales, Australia; Cat # D8900) containing 5% fetal calf serum (Sigma-Aldrich, New South Wales, Australia), 100 IU/ml penicillin, 0.01% streptomycin sulfate (GIBCO, Carlsbad, CA, USA; Cat # 15140122), 0.1% fungizone (GIBCO, Carlsbad, CA, USA. Cat # 15290–018) and 3 ng/ml human epidermal growth factor (EGF; Sigma-Aldrich, New South Wales, Australia; Cat # E9644). Total cell number was determined using trypan blue method and a Neubauer haemocytometer. Then 250 cells were seeded per wellin 24-well plates which were pre-coated with collagen type I (Sigma-Aldrich, New South Wales, Australia) at a density of 10 μg/cm$^2$. The cells were then incubated at 5% CO$_2$ and 38.5˚C. The culture medium was changed every two days and the growth of purified GREL or fibroblast clusters was checked every day by light microscopy (Olympus, Tokyo, Japan). Clusters with characteristic morphology were observed from day 2 onwards (S1 Fig). The majority of wells contained cultures of mixed cell types with fibroblast clusters and only one or two large purified GREL cell clusters in each 24-well plate. The largest purified and healthy clusters of GREL cells or fetal fibroblasts were selected on day 3 and day 4 (S1 Fig). In case the clusters were surrounded by some cells of another cell type (S1 Fig), these cells were discarded by scraping with a sterile blunt-ended glass Pasteur pipette and subsequent culture medium change. When the cell clusters were confluent (S2 and S3 Figs), GREL cells and fibroblasts were collected. To collect pure GREL cell or fibroblast cultures for subsequent RNA extraction, the clusters were scraped with a blunt-ended Pasteur pipette and the culture media with the cells transferred into sterile Eppendorf tubes. After centrifugation at 1500 rpm for 5 min, the cell pellets were stored at -80˚C until required.

### Isolation of adult ovarian fibroblasts

After collection, any connective tissue surrounding the adult ovaries was discarded and the ovaries washed with 70% ethanol and HBSS$^{+/+}$. Individual ovaries were cut into quarters and

tissue areas devoid of visible follicles was cut to a depth of about 0.5 mm from the surface and minced with sterile razor blades and digested with HBSS$^{+/+}$ containing 1 mg/ml collagenase type I and 50 μg/ml DNase I at 37˚C shaking at 100 rpm for 1 to 2 h. Subsequently, cell suspensions were passed through 70 μm cell strainers (Sartorius, Dandenong South, VIC, Australia) and centrifuged at 1500 rpm for 5 min. The cell pellet was resuspended in 5 ml 0.05% trypsin/ EDTA (GIBCO, Carlsbad, CA, USA/Life Technologies Australia Pty Ltd, Tullamarine, Vic, Australia) in HBSS without calcium or magnesium (HBSS$^{-/-}$) for digestion with pipetting, and the digestion was stopped with DMEM/F12 containing 5% fetal calf serum, 100 IU/ml penicillin, 0.01% streptomycin sulfate, 0.1% fungizone and 3 ng/ml EGF after 5 min. Total cell number was determined as described above for fetal cells and the adult fibroblasts seeded into 24-well plates and cultured in the same medium and way as the fetal ovarian cells. Purified clusters were collected and cell pellets were stored at -80˚C for subsequent use.

## Collection of adult ovarian surface epithelial cells

Individual adult ovaries were placed into a 60 mm culture dish and the connective tissues around the ovary was removed with forceps and scissors. Subsequently, ovaries were moved to another dish with 12 ml DMEM/F12 containing 5% fetal calf serum, 100 IU/ml penicillin, 0.01% streptomycin sulfate and 0.1% fungizone and the surface of each ovary was scraped with the blunt side of a scalpel blade. Ovarian surface epithelium from five to seven ovaries were pooled and transferred into a 15 ml centrifuge tube and centrifuged at 1500 rpm for 5 min. Supernatant was discarded and cell pellets were stored at -80˚C until required. In total 5 pools were collected.

## Collection of adult granulosa cells

Granulosa cells were isolated from the largest healthy antral follicle (9–12 mm) from each adult ovarian pair. The diameters of the follicles were measured with callipers. The follicles were cut out with scissors and transferred to a dish containing 1 ml HBSS$^{-/-}$. The follicles were cut open with scissors and granulosa cells scraped from the follicle wall with a blunt-ended Pasteur pipette. The cell suspension from each individual follicle was transferred into a 15 ml tube containing HBSS$^{-/-}$ and centrifuged at 1500 rpm for 5 min. Then the cell pellets were resuspended in 1 ml HBSS$^{-/-}$ and the ratio of alive/dead cells determined using the trypan blue method and a Neubauer haemocytometer. Cell suspensions containing more than 70% live cells were centrifuged at 1500 rpm for 5 min, and cell pellets stored at -80˚C for subsequent use.

## Treatment of fetal fibroblasts

Fetal fibroblasts were from a previous study and had been cultured and treated as described previously [26]. Note that these cells were not cultured with EGF during their isolation culture. Briefly, bovine fetal fibroblasts were seeded in 24-well plates at a density of 30,000 cells/well in DMEM/F12 containing 5% fetal calf serum, 1% penicillin and streptomycin sulphate and 0.1% fungizone. When cells were 60–70% confluent, cells were treated with different agents for 18 h. After 18 h, cells were harvested and stored at -80˚C. Expression of *NR2F2*, *FBN1*, *COL1A1* and *COL3A1* was subsequently measured. Cells that were treated with 24 different agents as listed in S1 Table were from weeks 13, 14, 17, 19 and 33 of gestation (n = 5) and those treated with TGFβ were from gestational weeks 19–26 (n = 6).

## RNA isolation, cDNA synthesis and quantitative real-time PCR

RNA from granulosa cells, ovarian surface epithelial cells and adult fibroblasts was extracted with TRIzol$^{TM}$ (Thermo Fisher Scientific, Waltham, MA, USA) according to manufacturer's

instructions. RNA from GREL cells and fetal fibroblasts was extracted with RNAqueous[TM]-Micro Total RNA Isolation kit. Ten μg of RNA of each sample was treated with DNase I (Promega Australia, Alexandria, NSW, Australia/Life Technologies Australia Pty Ltd, Tullamarine, Vic, Australia) for 20 min at 37°C. Two hundred ng of DNAse-treated RNA was used for cDNA synthesis as previously described [8, 29]. Based on available RNA sequences in NCBI, PCR primers were designed to span introns using Primer3 plus and Net primer software (PREMIER Biosoft Palo Alto, CA, USA) and primers are listed in S2 Table. Primer combinations were tested as previously described [29]. No addition of reverse transcriptase was used to confirm no amplification of any potential genomic DNA contamination. The amplification conditions were 95°C for 15 sec, then 60°C for 60 sec for 40 cycles using Rotor Gene 6000 cycler (Q series, Qiagen GmbH, Hilden, Germany). Ct values were determined using Rotor Gene 6000 software at a threshold of 0.05 normalised fluorescent unit. Gene expression values were determined using $2^{-\Delta Ct}$ method and ribosomal protein L32 (*RPL32*) and peptidylprolyl isomerase A (*PPIA*) were used as housekeeping genes. The fold change of gene expression in fetal fibroblasts treated with different agents were presented as $2^{-\Delta\Delta Ct}$ data with the same housekeeping genes in $2^{-\Delta Ct}$ method. We analysed the expression levels of 34 genes relating to cell junctions, extracellular matrix, enzymes and transcription factors in order to characterise the cultured GREL cells and fetal fibroblasts.

## Statistical analysis

All statistical analyses were carried out using Microsoft Office Excel 365 (Microsoft Redmond, WA, USA) and IBM SPSS Statistics for Windows, version 25 (IBM Corp., Armonk, N.Y., USA). Scatter plots and columns were made using GraphPad Prism version 8.00 (GraphPad Software Inc., La Jolla, CA, USA). Pearson correlation test was used to analyse the correlation between gene expression levels and gestational age. The differences between GREL cells and fetal fibroblasts, the differences between fetal fibroblasts and adult fibroblasts, and the differences between fetal fibroblasts cultured with or without EGF were analysed with Student's t-test. To determine significant differences among adult fibroblasts, granulosa cells and ovarian surface epithelial cells, and differences between GREL cells with adult ovarian surface epithelial cells or granulosa cells, one-way ANOVA and Tukey's post-hoc tests were conducted. To compare the difference between the control and the treatments in fetal fibroblasts, one-way ANOVA with Dunnett's post-hoc tests were conducted.

## Results

### Morphology of GREL cells and fetal fibroblasts in culture

Early in culture (day 2 and 3), GREL cells in clusters appeared to be of irregularly polygonal shape, with clear visible nuclei, and in some areas tightly packed (S1A and S1C Fig), as previously observed [8]. In the following days, cells appeared to be more tightly connected (S1E and S1F Fig) and remained so until the cells were ready for harvesting (Fig 1A and S2 Fig). In contrast, fetal fibroblasts showed the typical spindle shaped appearance as also seen in adult fibroblasts (Fig 1B, S1B and S1D Fig, S3 Fig). The purified GREL and fetal fibroblast clusters demonstrated that the current method of isolating GREL and fetal fibroblast was feasible. In order to explore more characters of GREL cells and fetal fibroblasts and to make comparisons with their adult counterparts, we conducted analysis of expression of genes including those involved in cellular junctions, extracellular matrix, hormone production, transcription factors, oestrogen and TGFβ signalling, cell cycle and stem and germ cells.

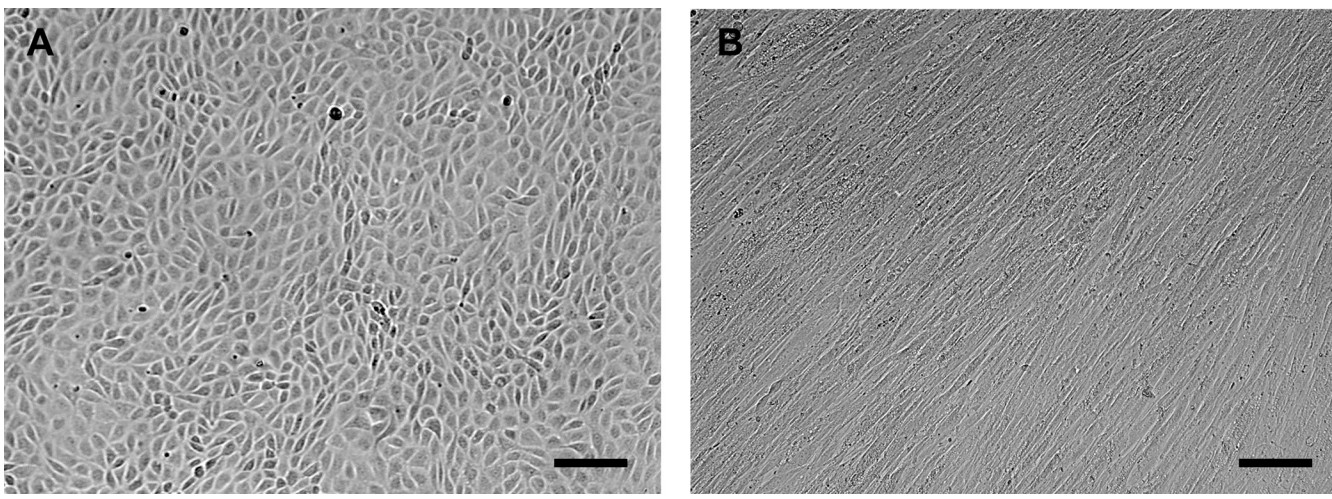

**Fig 1. Morphology of fetal somatic cells.** Representative micrograph image of (A) GREL cells and (B) fetal fibroblasts. The cell types were isolated from fetal ovaries with gestational ages of 56 days and 73 days, respectively. Bars: 100 μm.

## Presentation of gene expression data

In Figs 2 to 7 we present expression data of the 34 genes examined and then group them based upon the functionally of genes for each Figure. For each gene the expression across gestation is presented, and for 24 of the 34 genes this had been published previously and is acknowledged and reprinted in the current Figs for ease of comparison. The data from the other 10 genes are new data presented in this manuscript. We then present the expression in fetal fibroblasts and GREL cells and the statistical comparison between them. We then present the expression in adult fibroblasts, granulosa cells and ovarian surface epithelial cells and the statistical comparison between them. The data for each gene from gestation and from isolated cells in Figs 2 to 7 is plotted on the same scale for direct comparisons. The mean levels of expression in isolated cells and statistical comparisons of fetal and adult cells in presented in Table 1. The individual data of expression in fetal fibroblasts and GREL cells across gestation is presented in a supplementary figure. The data with correlations between the 10 new genes across gestation and age of gestation is presented in a supplementary table and cannot be combined with previously published data on the other 24 genes as the RNA used for the 10 new genes is from a different collection of fetal ovaries.

## Cellular junctions

In the fetal ovary, *KRT19* (Fig 2A), *OCLN* (Fig 2D and S3 Table), *CTNNB1* and (Fig 2G) were expressed constantly throughout the gestation. Expression of *DSG2* (Fig 2B), *PKP2* (Fig 2C and S3 Table) and *MUC1* (Fig 2E and S3 Table) were negatively correlated with gestational age, however, expression of *GJA1* (Fig 2F and S3 Table) was positively correlated with gestational age. In GREL cells and fetal fibroblasts, expression of *KRT19* (Fig 2A), *PKP2* (Fig 2C), *OCLN* (Fig 2D) and *MUC1* (Fig 2E) was significantly higher in GREL cells than fetal fibroblasts. *DSG2* (Fig 2B) did not exhibit significant differences between GREL cells and fetal fibroblasts, even though most GREL cells exhibited higher *DSG2* expression (S4A2 Fig) than fetal fibroblasts from the same fetus. *GJA1* (Fig 2F) had significantly higher expression in fetal fibroblasts. Expression of *CTNNB1* (Fig 2G) did not exhibit significant differences ($P > 0.05$) between GREL cells and fetal fibroblasts. In adult cells, expression of *KRT19* (Fig 2A), *PKP2* (Fig 2C), *OCLN* (Fig 2D) and *MUC1* (Fig 2E) was greater in ovarian surface epithelial cells,

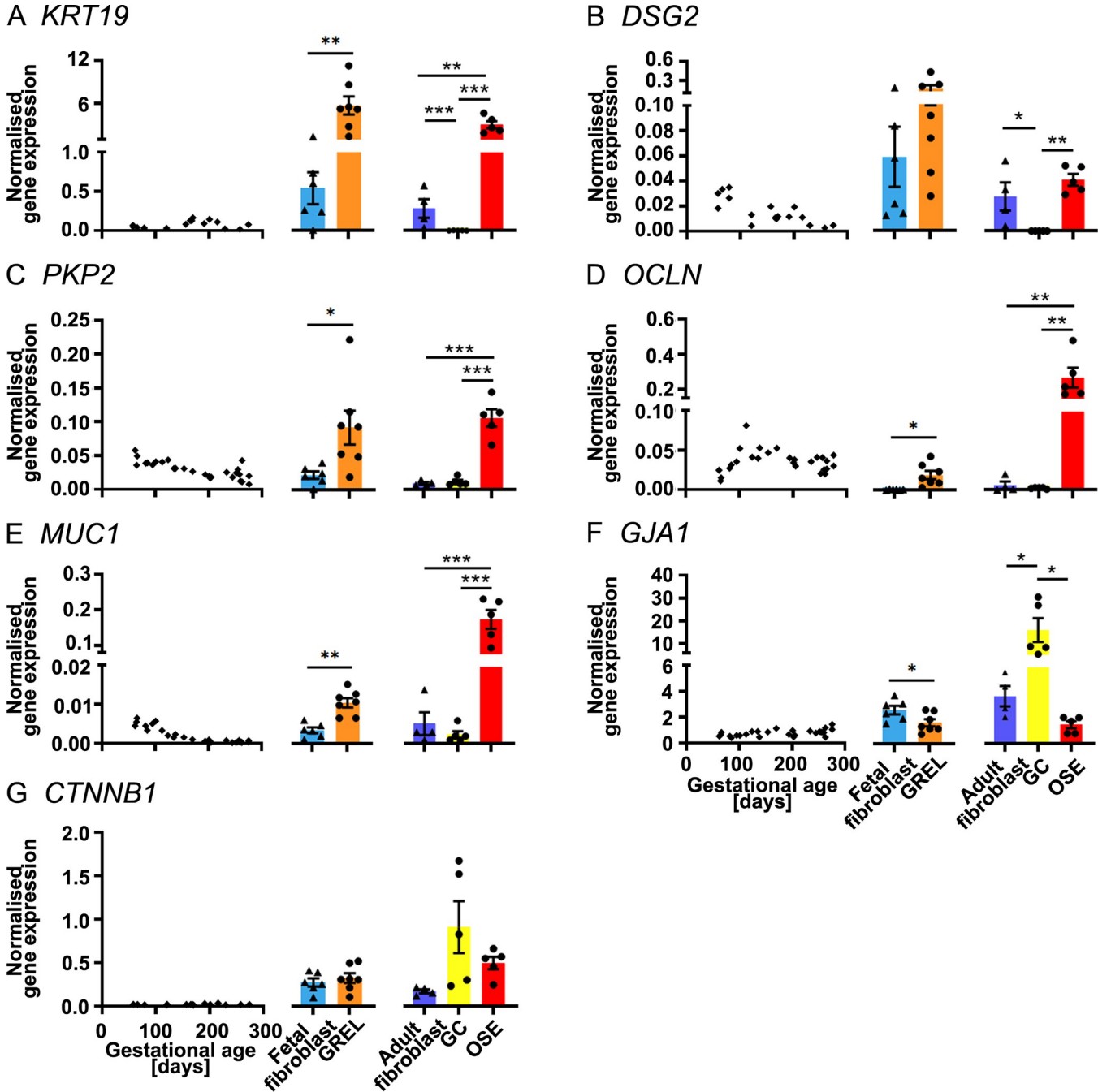

**Fig 2. mRNA expression levels of genes associated with cellular junctions.** Scatter plots in each subfigure show gene expression levels in fetal ovaries at different gestational ages (n = 27), and scatter plots for *KRT19* (A), *DSG2* (B) and *CTNNB1* (G) are from our previous publications [30, 31]. Gene expression levels in cultured GREL cells (n = 7; gestational ages are 51, 56, 73, 110, 110, 127 and 177 days) and fetal fibroblasts (n = 6; gestational ages are 51, 73, 101, 110, 110 and 177 days) are presented as mean ± SEM (normalised to *PPIA* and *RPL32*) in the middle graph of each subfigure. The differences between GREL cells and fetal fibroblasts were analysed with Student's t-test. Data of gene expression in adult fibroblasts (n = 4), granulosa cells (GC; n = 5) and ovarian surface epithelium (OSE; n = 5) are presented as mean ± SEM (normalised to *PPIA* and *RPL32*) in the graph on the right side of each subfigure. Significant differences among the three adult cell types were determined by one-way ANOVA with Tukey's post-hoc test. * $P < 0.05$, ** $P < 0.01$, *** $P < 0.001$.

whereas *GJA1* (Fig 2F) was higher in granulosa cells. *DSG2* (Fig 2B) was more highly expressed in ovarian surface epithelial cells and adult fibroblasts than in granulosa cells. Additionally expression of *CTNNB1* (Fig 2G) was not significantly different between the three cell types.

**Table 1. Relative gene expression levels (2-ΔCt) are presented as mean ± SEM for each cell type.** Comparisons are between fetal fibroblasts (n = 6) with adult fibroblasts (n = 4), and then fetal GREL cells (n = 7) with adult ovarian surface epithelial cells (n = 5) or granulosa cells (n = 5).

| Gene | Fetal fibroblasts | Adult fibroblasts | Fetal GREL cells | Adult ovarian surface epithelial cells | Adult granulosa cells |
|---|---|---|---|---|---|
| KRT19 | 0.542±0.203 | 0.286±0.120 | 5.044±1.313 | 3.090±0.515 | 0.003±0.001** |
| DSG2 | 0.059±0.024 | 0.027±0.011 | 0.132±0.048 | 0.041±0.005 | 0.000±0.000* |
| PKP2 | 0.022±0.005 | 0.01±0.002 | 0.080±0.024 | 0.106±0.013 | 0.012±0.003*** |
| OCLN | 0.001±0.000 | 0.006±0.005 | 0.017±0.005 | 0.272±0.057*** | 0.003±0.000** |
| MUC1 | 0.004±0.001 | 0.005±0.003 | 0.009±0.002 | 0.180±0.027*** | 0.002±0.001*** |
| GJA1 | 2.640±0.332 | 3.732±0.798 | 1.461±0.319 | 1.507±0.287 | 17.049±5.306** |
| CTNNB1 | 0.281±0.047 | 0.177±0.023 | 0.290±0.064 | 0.504±0.071 | 0.922±0.302* |
| LAMB2 | 0.193±0.026 | 0.130±0.037 | 0.165±0.037 | 0.856±0.191** | 0.650±0.107* |
| FBN1 | 0.913±0.103 | 1.425±0.161* | 0.374±0.104 | 0.211±0.029 | 0.012±0.003** |
| COL1A1 | 16.111±2.289 | 14.009±2.382 | 4.653±1.984 | 3.907±0.640 | 0.043±0.021*** |
| COL3A1 | 6.116±1.553 | 4.484±0.453 | 3.613±1.339 | 2.186±0.381 | 0.009±0.003* |
| COL4A1 | 1.797±0.845 | 0.329±0.081 | 0.752±0.418 | 0.301±0.129 | 0.441±0.127 |
| NR5A1 | 0.061±0.009 | 0.028±0.006* | 0.016±0.007 | 0.120±0.021** | 0.416±0.105*** |
| STAR | 0.038±0.011 | 0.007±0.002 | 0.018±0.005 | 0.009±0.002 | 0.002±0.001*** |
| CYP11A1 | 0.005±0.002 | 0.039±0.017 | 0.001±0.001 | 0.127±0.077*** | 3.758±1.128*** |
| HSD3B1 | 0.000±0.000 | 0.000±0.000 | 0.000±0.000 | 0.005±0.002*** | 0.316±0.116*** |
| CYP19A1 | 0.001±0.000 | 0.000±0.000 | 0.000±0.000 | 0.056±0.051* | 10.307±2.106*** |
| HSD17B1 | 0.000±0.000 | 0.000±0.000 | 0.000±0.000 | 0.003±0.001*** | 0.328±0.157*** |
| INHBA | 0.002±0.000 | 0.001±0.000 | 0.001±0.000 | 0.008±0.005 | 0.005±0.002 |
| NR2F2 | 0.559±0.128 | 0.455±0.096 | 0.135±0.037 | 0.172±0.024 | 0.050±0.016 |
| FOXL2 | 0.051±0.008 | 0.024±0.006* | 0.012±0.006 | 0.028±0.006 | 1.015±0.280*** |
| ESR1 | 0.001±0.001 | 0.002±0.001 | 0.014±0.007 | 0.570±0.094** | 0.116±0.010* |
| ESR2 | 0.007±0.001 | 0.003±0.000** | 0.009±0.006 | 0.022±0.003* | 0.144±0.039*** |
| TGFBR1 | 0.148±0.033 | 0.06±0.008* | 0.153±0.036 | 0.136±0.013 | 0.088±0.015 |
| TGFBR2 | 0.105±0.026 | 0.05±0.011 | 0.129±0.037 | 0.533±0.076* | 0.007±0.006*** |
| TGFBR3 | 0.195±0.058 | 0.065±0.038 | 0.125±0.027 | 0.483±0.096** | 0.325±0.083* |
| CCND2 | 0.096±0.011 | 0.018±0.003** | 0.038±0.011 | 0.111±0.032 | 0.480±0.156** |
| CCNE1 | 0.038±0.005 | 0.009±0.002** | 0.018±0.004 | 0.025±0.011 | 0.024±0.006 |
| CCNE2 | 0.039±0.005 | 0.005±0.001** | 0.024±0.005 | 0.022±0.007 | 0.060±0.0170 |
| ALDH1A1 | 0.095±0.026 | 0.102±0.030 | 0.019±0.003 | 0.038±0.011 | 0.008±0.002 |
| LGR5 | 0.000±0.000 | 0.000±0.000 | 0.001±0.000 | 0.045±0.007* | 0.000±0.000 |
| OCT4 | 0.012±0.001 | 0.007±0.001* | 0.011±0.003 | 0.039±0.006** | 0.014±0.003 |
| DAZL | 0.003±0.001 | 0.001±0.000 | 0.001±0.000 | 0.003±0.001 | 0.001±0.000 |
| VASA | 0.001±0.000 | 0.000±0.000* | 0.001±0.000 | 0.001±0.001 | 0.000±0.000 |

Student's t-tests were used to analyse the differences between fetal fibroblasts and adult fibroblasts. One-way ANOVA with Tukey's post-hoc test was used to analyse the differences between GREL cells with adult ovarian surface epithelial cells or granulosa cells

\* $P < 0.05$

\*\* $P < 0.01$

\*\*\* $P < 0.001$ report significant differences between the corresponding fetal cells with the adult cells.

## Extracellular matrix

In the fetal ovary, expression of *LAMB2* (Fig 3B and S3 Table), *FBN1* (Fig 3C), *COL1A1* (Fig 3D), *COL3A1* (Fig 3E) and *COL4A1* (Fig 3F and S3 Table) were positively correlated with the gestational age. *FBN1* (Fig 3C) and *COL1A1* (Fig 3D) levels were significantly higher in fetal fibroblasts than in GREL cells. *LAMB2* (Fig 3B), *COL3A1* (Fig 3E) and *COL4A1* (Fig 3F)

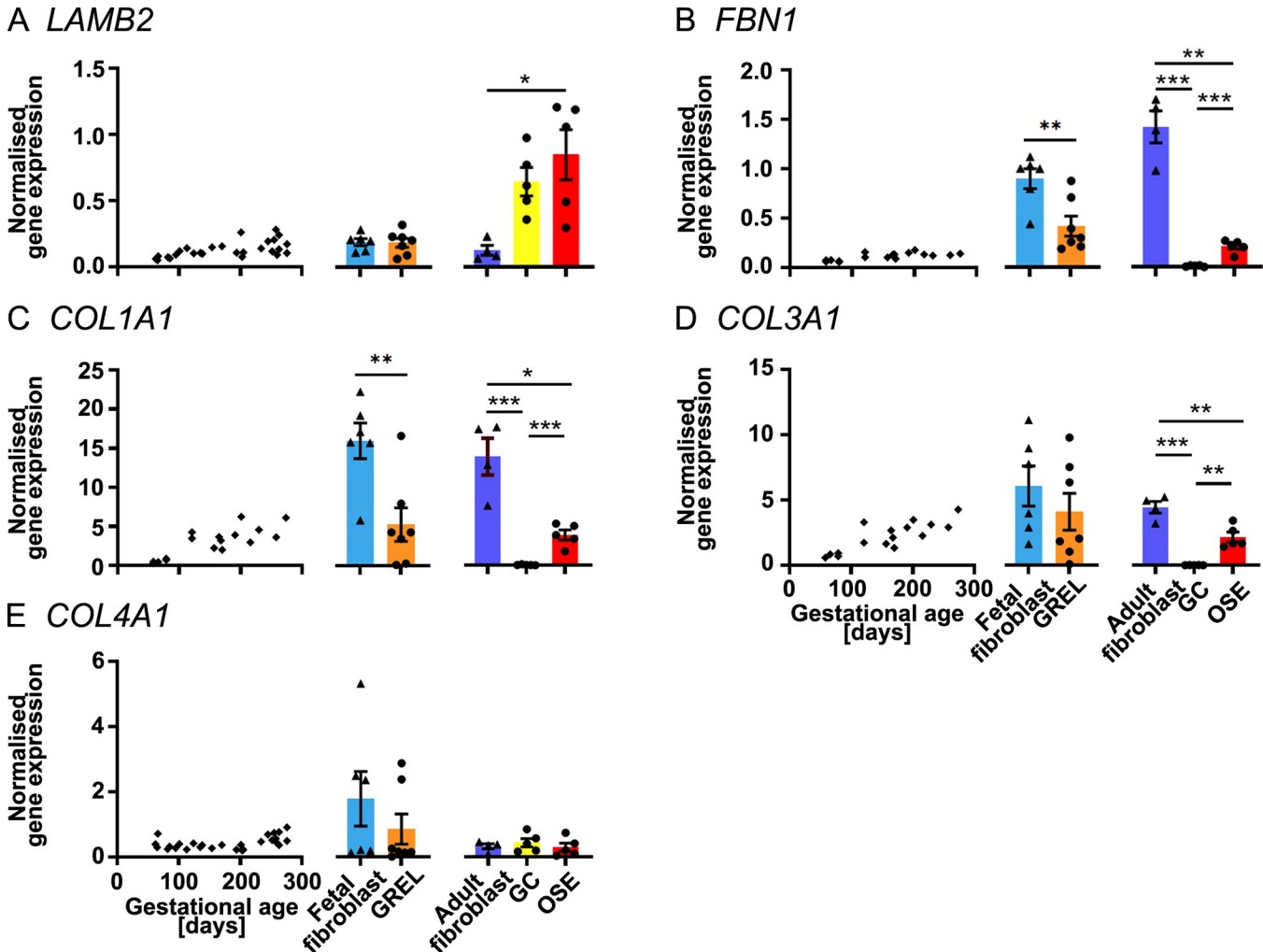

**Fig 3. mRNA expression levels of extracellular matrix genes.** Scatter plots in each subfigure show gene expression levels in fetal ovaries at different gestational ages (n = 27), and scatter plots for *FBN1* (B), *COL1A1* (C) and *COL3A1* (D) are from our previous publications [30, 31]. Gene expression levels in cultured GREL cells (n = 7; gestational ages are 51, 56, 73, 110, 110, 127 and 177 days) and fetal fibroblasts (n = 6; gestational ages are 51, 73, 101, 110, 110 and 177 days) are presented as mean ± SEM (normalised to *PPIA* and *RPL32*) in the middle graph of each subfigure. The differences between GREL cells and fetal fibroblasts were analysed with Student's t-tests. Data of gene expression in adult fibroblasts (n = 4), granulosa cells (GC; n = 5) and ovarian surface epithelium (OSE; n = 5) are presented as mean ± SEM (normalised to *PPIA* and *RPL32*) in the graph on the right side of each subfigure. Significant differences among the three adult cell types were determined by one-way ANOVA with Tukey's post-hoc test. * $P < 0.05$, ** $P < 0.01$, *** $P < 0.001$.

expression did not exhibit significant differences between GREL cells and fetal fibroblasts. In adult cells, adult fibroblasts had significant higher expression of *FBN1* (Fig 3C) and *COL1A1* (Fig 3D) and significantly lower expression of *LAMB2* (Fig 3B) than ovarian surface epithelial cells and granulosa cells. Ovarian epithelial cells and adult fibroblast had higher expression of *COL3A1* (Fig 3E) than granulosa cells. *COL4A1* (Fig 3F) did not have significant differences among the three cell types.

## Genes associated with hormone production

In the fetal ovary, expression levels of *NR5A1* (Fig 4A), *CYP11A1* (Fig 4C) and *HSD3B1* (Fig 3D) were not significantly correlated with gestational age. *STAR* (Fig 4B and S3 Table), *HSD17B1* (Fig 4F and S3 Table) and *INHBA* (Fig 4G) were positively correlated with gestational age but *CYP19A1* (Fig 4E) was negatively correlated with gestational age. Fetal

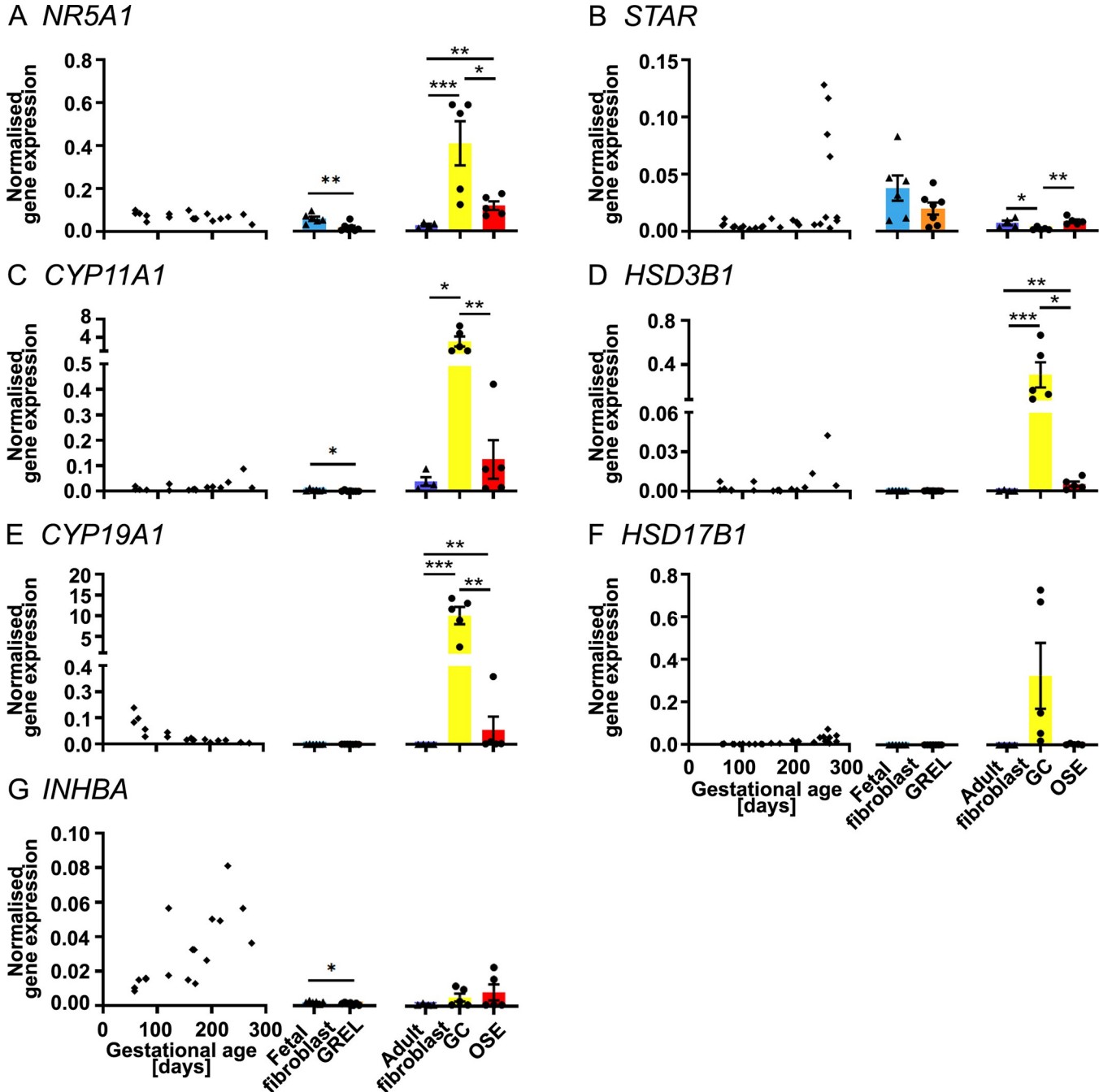

**Fig 4. mRNA expression levels of genes involved in hormone production.** Scatter plots in each subfigure show gene expression levels in fetal ovaries at different gestational ages (n = 27), and scatter plots for *NR5A1* (A), *CYP11A1*(C), *HSD3B1* (D), *CYP19A1* (E) and *INHBA* (G) are from our previous publications [30, 31]. Gene expression levels in cultured GREL cells (n = 7; gestational ages are 51, 56, 73, 110, 110, 127 and 177 days) and fetal fibroblasts (n = 6; gestational ages are 51, 73, 101, 110, 110 and 177 days) are presented as mean ± SEM (normalised to *PPIA* and *RPL32*) in the middle graph of each subfigure. The differences between GREL cells and fetal fibroblasts were analysed with Student's t-tests. Data of gene expression in adult fibroblasts (n = 4), granulosa cells (GC; n = 5) and ovarian surface epithelium (OSE; n = 5) are presented as mean ± SEM (normalised to *PPIA* and *RPL32*) in the graph on the right side of each subfigure. Significant differences among the three adult cell types were determined by one-way ANOVA with Tukey's post-hoc test. * $P < 0.05$, *** $P < 0.001$.

fibroblasts had higher expression of *NR5A1* (Fig 4A) than GREL cells. Likewise, *CYP11A1* (Fig 4C) and *INHBA* (Fig 4G) had also a significantly higher expression in fetal fibroblasts, even though it was very low. *STAR* did not show significance differences between GREL cells and fetal fibroblasts (Fig 4B), even if most GREL cells had higher *STAR* expression than fetal fibroblasts of the same age (S4C2 Fig). Expression of *HSD3B1* (Fig 4D), *CYP19A1* (Fig 4E) and *HSD17B1* (Fig 4F) exhibited no significant differences between GREL cells and fetal fibroblasts, and they were expressed at very low levels. In adult cells, *CYP11A1* (Fig 4C), *HSD3B1* (Fig 4D) and *CYP19A1* (Fig 4E) were more highly expressed in granulosa cells. *NR5A1* (Fig 3A) was more highly expressed in both granulosa cells and ovarian surface epithelium than adult fibroblasts. Ovarian surface epithelial cells and adult fibroblast had higher expression of *STAR* (Fig 4B) than granulosa cells. *HSD17B1* (Fig 4F) and *INHBA* (Fig 4G) did not exhibit significant differences among the three cell types.

## Transcription factors, oestrogen receptors and TGFβ signalling pathways

In the fetal ovary, *NR2F2* (Fig 5A), *ESR1* (Fig 5C), *TGFBR2* (Fig 5F) and *TGFBR3* (Fig 5G) were positively correlated with gestational age but *ESR2* (Fig 5D) was negatively correlated with gestational age. *TGFBR1* (Fig 5E) and *FOXL2* (Fig 5B) were expressed constantly during the gestation. In GREL cells and fetal fibroblasts, *ESR1* (Fig 5C) was expressed significantly higher in GREL cells. *ESR2* (Fig 5D) was hardly detectable in GREL cells or fetal fibroblasts. *FOXL2* (Fig 5B) and *NR2F2* (Fig 5A) was expressed significantly higher in fetal fibroblasts. GREL cells and fetal fibroblasts did not exhibit significant differences in expression of *TGFBR1* (Fig 5E), *TGFBR2* (Fig 5F) and *TGFBR3* (Fig 5G). In adult cells, *ESR1* (Fig 5C), *TGFBR1* (Fig 5E) and *TGFBR2* (Fig 5F) were more highly expressed in ovarian surface epithelial cells and *ESR2* (Fig 5D) and *FOXL2* (Fig 5B) were more highly expressed in granulosa cells. *TGFBR3* (Fig 5G) was expressed more in both surface epithelial cells and granulosa cells than in adult fibroblasts. Adult fibroblasts had significant higher expression of *NR2F2* (Fig 5A) than ovarian surface epithelial cells and granulosa cells.

## Cell cycle genes

In the fetal ovary, expression of *CCND2* (Fig 6A), *CCNE1* (Fig 6B and S3 Table) and *CCNE2* (Fig 6C and S3 Table) was negatively correlated with gestational age. CCNE1 and CCNE2 levels reached to a peak around 100 days of gestation before declining. In GREL cells and fetal fibroblasts, expression of *CCND2* (Fig 6A) and *CCNE1* (Fig 6B) was significantly higher in fetal fibroblasts, but *CCNE2* (Fig 6C) did not exhibit significant differences between the two cell types. In adult cells, expression of *CCND2* (Fig 6A) was higher in granulosa cells. *CCNE2* (Fig 6C) was more highly expressed in both granulosa cells and ovarian surface epithelium than in adult fibroblasts, however, there was no statistically significant difference between adult ovarian surface epithelial cells and adult fibroblasts. Expression of *CCNE1* (Fig 6B) was not significantly difference among the three adult cell types.

## Stem or germ cell genes

In the fetal ovary, expression of *ALDH1A1* (Fig 7A) was positively correlated with gestational age but *OCT4* (Fig 7C) was negatively correlated with gestational age. *LGR5* (Fig 7B), *DAZL* (Fig 7D) and *VASA* (Fig 7E) were constantly expressed throughout the gestation. In GREL cells and fetal fibroblasts, expression of *ALDH1A1* (Fig 7A) was significantly higher in fetal fibroblasts but *LGR5* (Fig 7B) was significantly higher in GREL cells. *OCT4* (Fig 7C), *DAZL* (Fig 7D) and *VASA* (Fig 7E) were hardly detected in GREL cells and fetal fibroblasts as expected. In adult cells, *LGR5* (Fig 7B) and *OCT4* (Fig 7C) had higher expression in the

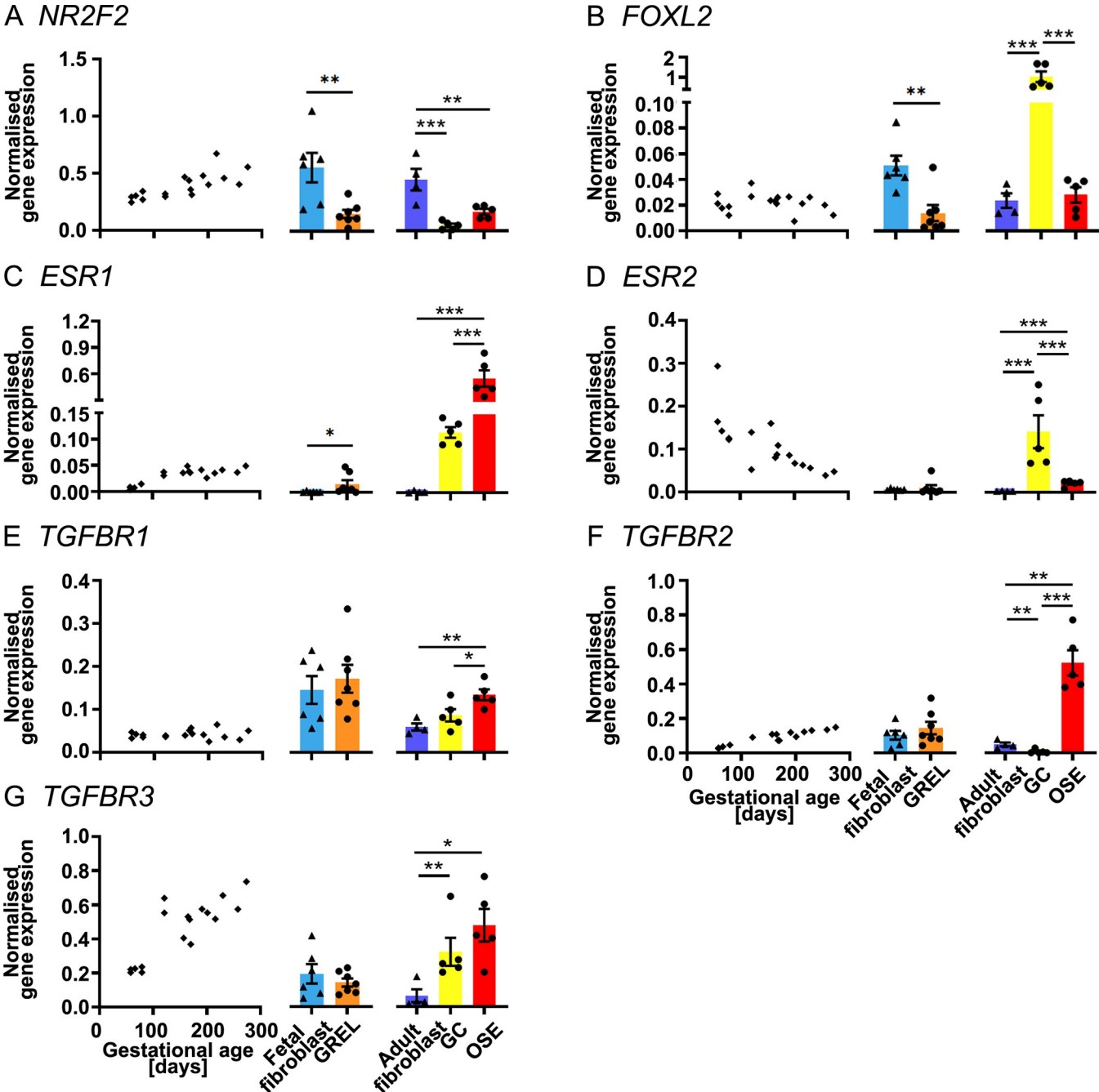

**Fig 5. mRNA expression levels of transcription factors, oestrogen receptors and TGFβ signalling pathways.** Scatter plots left in each subfigure of gene expression levels in fetal ovaries at different gestational ages (n = 27) are from our previous publications [30, 31]. Gene expression levels in cultured GREL cells (n = 7; gestational ages are 51, 56, 73, 110, 110, 127 and 177 days) and fetal fibroblasts (n = 6; gestational ages are 51, 73, 101, 110, 110 and 177 days) are presented as mean ± SEM (normalised to *PPIA* and *RPL32*) in the middle graph of each subfigure. The differences between GREL cells and fetal fibroblasts were analysed with Student's t-tests. Data of gene expression in adult fibroblasts (n = 4), granulosa cells (GC; n = 5) and ovarian surface epithelium (OSE; n = 5) are presented as mean ± SEM (normalised to *PPIA* and *RPL32*) in the graph on the right side of each subfigure. Significant differences among the three adult cell types were determined by one-way ANOVA with Tukey's post-hoc test. * $P < 0.05$, ** $P < 0.01$, *** $P < 0.001$.

ovarian surface epithelium. Similarly, *ALDH1A1* (Fig 7A) had higher expression in both ovarian surface epithelium and fibroblasts than granulosa cells. *DAZL* (Fig 7D) and *VASA* (Fig 7E) were hardly detectable in the three cell types.

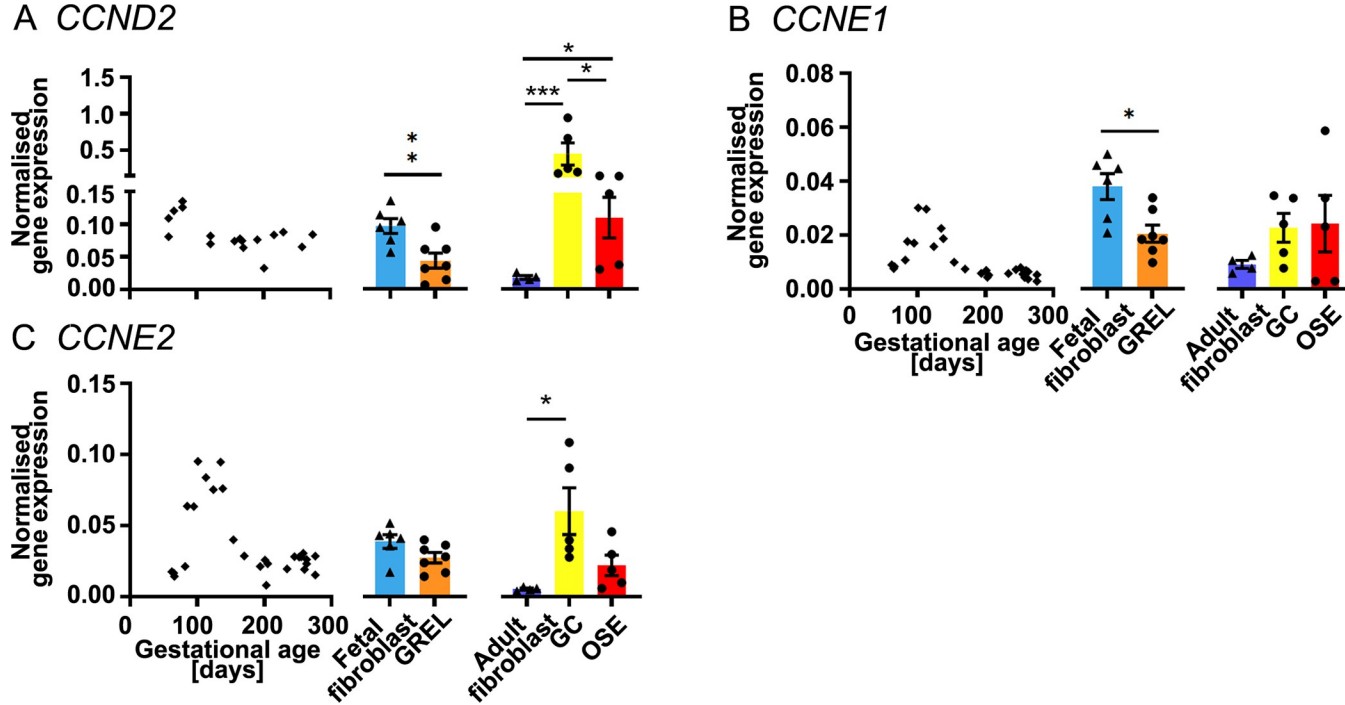

**Fig 6. mRNA expression levels of cell cycle genes.** Scatter plots in each subfigure show gene expression levels in fetal ovaries at different gestational ages (n = 27), and the scatter plot for *CCND2* (A) is from our previous publication [31]. Gene expression levels in cultured GREL cells (n = 7; gestational ages are 51, 56, 73, 110, 110, 127 and 177 days) and fetal fibroblasts (n = 6; gestational ages are 51, 73, 101, 110, 110 and 177 days) are presented as mean ± SEM (normalised to *PPIA* and *RPL32*) in the middle graph of each subfigure. The differences between GREL cells and fetal fibroblasts were analysed with Student's t-tests. Data of gene expression in adult fibroblasts (n = 4), granulosa cells (GC; n = 5) and ovarian surface epithelium (OSE; n = 5) are presented as mean ± SEM (normalised to *PPIA* and *RPL32*) in the graph on the right side of each subfigure. Significant differences among the three adult cell types were determined by one-way ANOVA with Tukey's post-hoc test. * $P < 0.05$, ** $P < 0.01$, *** $P < 0.001$.

### Fetal versus adult cells

*CCND2*, *CCNE1*, *CCNE2*, *ESR2* and *TGFBR1* were more highly expressed in the fetal fibroblasts than in adult fibroblasts (Table 1) whereas *FBN1* was lower in the fetal fibroblasts. Expression of *OCLN*, *MUC1*, *LAMB2*, *NR5A1*, *ESR1*, *ESR2* and *TGFBR3* was lower in GREL cells than ovarian surface epithelial cells (Table 1). Expression of *NR5A1*, *CYP11A1*, *HSD3B1*, *CYP19A1* and *HSD17B1* was statistically lower in GREL cells than ovarian surface epithelial cells but all were much lower than granulosa cells (Table 1). Expression of *KRT19*, *DSG2*, *PKP2*, *OCLN*, *MUC1*, *FBN1*, *COL1A1*, *COL3A1*, *STAR* and *TGFBR2* was higher in GREL cells compared to granulosa cells whereas expression of *GJA1*, *CTNNB1*, *LAMB2*, *NR5A1*, *CYP11A1*, *HSD3B1*, *CYP19A1*, *HSD17B1*, *FOXL2*, *ESR1*, *ESR2*, *TGFBR3* and *CCND2* was lower (Table 1).

### Regulation of genes in fetal fibroblasts

Fetal fibroblasts previously isolated and cultured without EGF [26] and treated with 24 of hormones, stimulators and growth factors were analysed for the expression of *NR2F2*, *FBN1*, *COL1A1* and *COL3A1* which are commonly considered as genes expressed by ovarian fibroblasts. Results showed that many of the treatments had no significant effect on the expression of those genes (S5 Fig). However, results of treatment of doses of TGFβ1 (5 or 20 ng/ml) on fetal fibroblasts (n = 6 from the second trimester of gestation) showed significantly increased *FBN1* expression in fetal fibroblast and significantly decreased the expression of *COL1A1* and *COL3A1* (Fig 8).

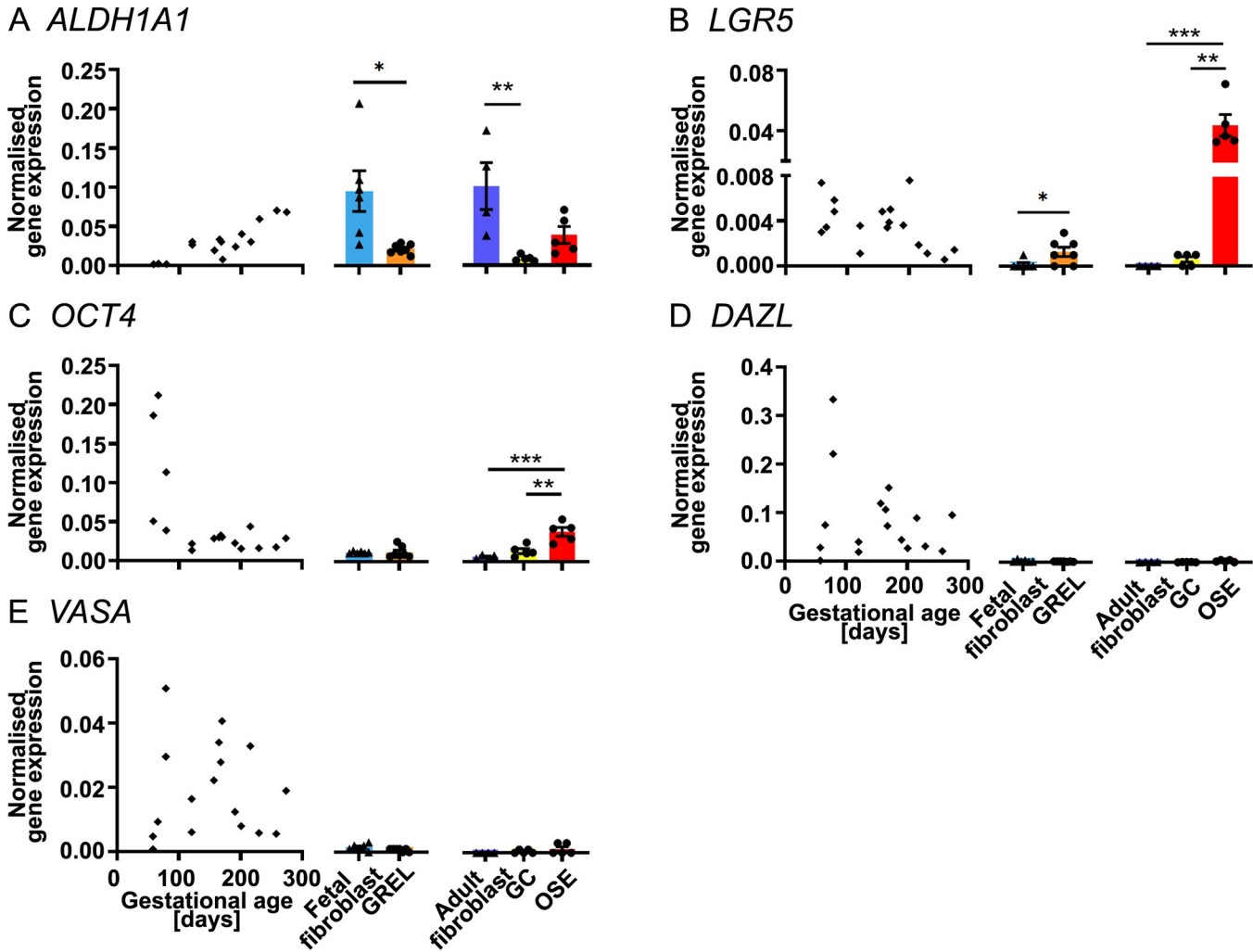

**Fig 7. mRNA expression levels of stem and germ cell genes.** Scatter plots in each subfigure show gene expression levels in fetal ovaries at different gestational ages (n = 27) and are from our previous publications [30, 31]. Gene expression levels in cultured GREL cells (n = 7; gestational ages are 51, 56, 73, 110, 110, 127 and 177 days) and fetal fibroblasts (n = 6; gestational ages are 51, 73, 101, 110, 110 and 177 days) are presented as mean ± SEM (normalised to *PPIA* and *RPL32*) in the middle graph of each subfigure. The differences between GREL cells and fetal fibroblasts were analysed with Student's t-tests. Data of gene expression in adult fibroblasts (n = 4), granulosa cells (GC; n = 5) and ovarian surface epithelium (OSE; n = 5) are presented as mean ± SEM (normalised to *PPIA* and *RPL32*) in the graph on the right side of each subfigure. Significant differences among the three adult cell types were determined by one-way ANOVA with Tukey's post-hoc test. * $P < 0.05$, ** $P < 0.01$, *** $P < 0.001$.

We compared the fetal fibroblast cultured in EGF-containing medium (n = 6 from 51, 56, 73, 110, 127 and 177 days of gestation) isolated in the current experiments with fetal fibroblasts cultured in EGF-free medium (n = 11 from the first, second and third trimester of gestation) from a previous study [26]. The result showed that EGF significantly increased *FBN1* expression, but decreased the gene expression of *COL1A1* (S6 Fig).

## Discussion

In this study we developed methods for isolating and culturing fetal ovarian GREL cells and fetal ovarian fibroblasts by removal of contaminating cells during culture and expansion from single colonies without passage. We had to adopt this approach because we had not identified a specific antigen on the surface of GREL cells to allow us to utilise more direct cell isolation

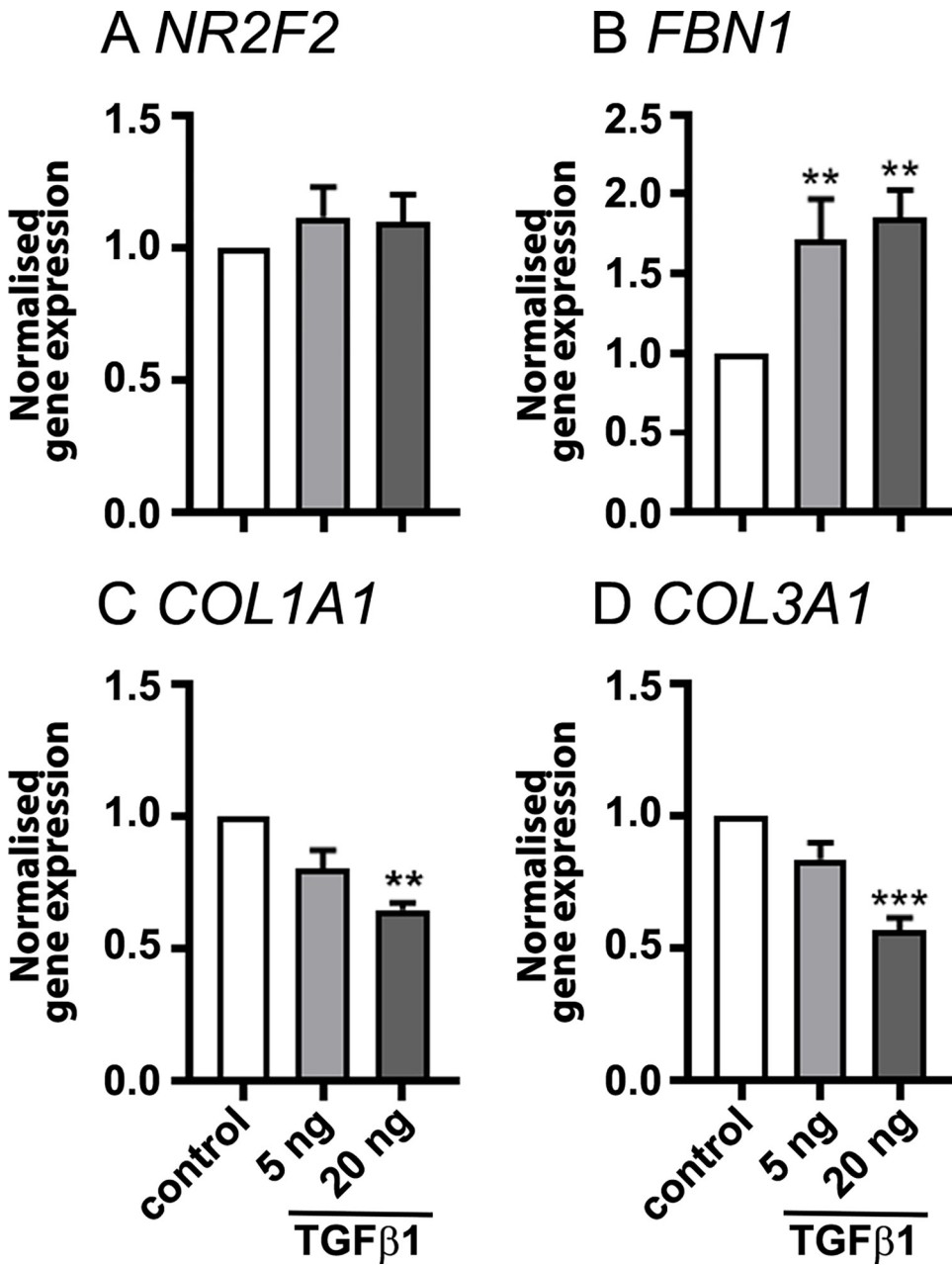

**Fig 8. Normalised gene expression genes in fetal fibroblasts.** Fibroblasts from bovine fetal ovaries (19–26 weeks, n = 6) were treated with 5 or 20 ng/ml TGFβ1 for 18 h. Fold changes of gene expression to the control groups are presented as mean ± SEM (normalised to *PPIA* and *RPL32*). Significant differences between groups were determined by one-way ANOVA with Tukey's post-hoc test. ** $P < 0.01$, *** $P < 0.001$.

methods such as the use of magnetic beads or fluorescence-activated cell sorting. We investigated the expression of a variety of genes to identify properties of GREL cells and fetal fibroblasts and examined differences with their adult counterparts. It should be acknowledged that culturing of GREL cells could have contributed to differences observed with the granulosa cells or ovarian surface epithelial cells which were not cultured. Furthermore, we additionally carried out a series of treatments to identify possible regulators of extracellular matrix in fetal fibroblasts. This is the first study of GREL cells since their identification.

Of the 10 newly-examined genes and the 24 previously-examined genes in whole fetal ovaries across gestation [30, 31] that we examined here in *in-vitro* cultured GREL cells and fetal fibroblasts, many were expressed early in gestation (*MUC1*, *PKP2*, *CCNE1*, *CCNE2*, *CCND2*, *CYP19A1*, *DSG2*, *ESR2* and *OCT4*) and were negatively correlated with gestational age. Others were expressed late in gestation (*ALDH1A1*, *COL1A1*, *COL3A1*, *ESR1*, *FBN1*, *INHBA*, *NR2F2*, *TGFBR2*, *TGFBR3*, *STAR*, *COL4A1*, *GJA1*, *LAMB2* and *HSD17B1*) and were positively correlated with gestational age. The expression of *NR2F2*, *FBN1*, *COL1A1* and *COL3A1* that were all positively correlated with the gestational age, could be as a result of the increasing volume of stroma in the fetal ovary [32]. Many of the 10 newly-examined genes were either negatively or positively correlated with each other. These patterns of gene expression and correlations between genes have been observed previously [28–31] and reinforce the major developmental processes that the ovary undergoes during its development [8].

Many earlier studies of fetal ovaries focused on germ cells and often refer to the other cells as somatic cells which to some degree creates the impression that they are one cell type. However, our earlier studies identified that stromal fibroblasts and GREL cells have different cellular origins and exist in the ovary in different cellular compartments separated by a basal lamina [8]. In the current study the morphology of the cells, when cultured, differed substantially. The fibroblasts had a classical elongated shape and the GREL cells were of irregular polygonal shape and were tightly packed together in colonies. Their gene expression profiles were also different. Fifteen out of the 34 genes examined were differentially expressed with GREL cells having higher expression of *KRT19*, *PKP2*, *OCLN*, *MUC1*, *ESR1* and *LGR5* and lower expression of *GJA1*, *FOXL2*, *NR2F2*, *FBN1*, *COL1A1*, *NR5A1*, *CCND2*, *CCNE1* and *ALDH1A1*. That *FOXL2* was higher in the fibroblasts is at odds with our previous immunohistochemistry data [8]. Our previous microarray data demonstrated that GREL cells in culture had higher signal intensity of *KRT19*, *DSG2* and *PKP2* than cultured fibroblasts isolated from the adult ovary [8] and this is consistent our current on *KRT19* and *PKP*2. There was a trend for higher *DSG2* expression in GREL cells, but not significantly so. The elevated levels of *NR2F2*, *FBN1* and *COL1A1* in stromal fibroblasts is in agreement with previous immunolocalisation studies [8, 32]. Thus fetal fibroblasts and GREL cells are clearly different somatic cell types.

During development of the ovary the GREL cells, being the only somatic cells on the surface, mature into surface epithelial cells once the stroma has penetrated to below the surface and spread laterally below it [8, 9]. It is when the stroma underlies the cells on the surface that GREL cells form into a mature epithelium overlaying an epithelial basal lamina that separates the epithelium from the underlying stroma. The position of basal lamina and stroma on the basal side of the surface epithelial cells is important for the apical-basal polarity of epithelia and much has been written about the effects of stroma on adjacent epithelial cells both in organ development and in diseases processes such as cancer [33]. From the current study, the cells in the adult ovary most closely related in phenotype to the GREL cells were the surface epithelial cells. However, the adult surface epithelial cells had higher expression of *OCLN*, *MUC1*, *LAMB2*, *NR5A1*, *ESR1*, *ESR2* and *TGFBR3*. *DSG2*, *KRT19* and *PKP2* were also highly expressed in GREL cells and ovarian surface epithelial cells. It has been found that cytokeratins, desmogleins and plakophilins bind to E-cadherin to facilitate formation of desmosomes [34–36], suggesting that desmosomes exist in both, GREL cells and the ovarian surface epithelium. The higher expression of *OCLN* suggests that the epithelium on the surface of the adult ovary has matured with tight junctions between the cells and the higher expression of *MUC1* suggests that they are actively and directionally secreting mucins. Mucins are expressed in epithelia contacting relatively harsh microenvironments and form a mucin layer on the apical surface of those epithelia [37, 38]. As such the adult surface epithelium would have apical-basal

polarity cued by its interface with underlying stroma. The GREL cells in the developing ovary would not have developed that polarity until such time as the stroma has completed its penetration and expanded laterally below the surface. This is completed by the end of gestation [32].

GREL cells also have the bi-potential ability and develop into granulosa cells of the follicles [8]. They are the only somatic cells in the ovigerous cords. The other somatic cells of the ovary are outside the cords in stroma and are separated from the inside of the cords by a basal lamina. In the current study there were many genes that differed between the GREL cells and granulosa cells suggesting that the GREL cells would need to change considerably to form granulosa cells. It would not just be a process of gaining functions as GREL cells would need to reduce expression of *KRT19*, *DSG2* and *PKP2* and thus differentiating into granulosa cells may be more complex than the differentiation into ovarian surface epithelium. However, this might possibly not be the case as the comparison we made was with granulosa cells from mature antral follicles. By this stage the granulosa cells are replicating and forming a multilayered membrana granulosa and are hormonally active. In fact five of the genes differentially expressed between GREL cells and mature granulosa cells have been previously identified to be differentially expressed (*COL4A1*, *INHBA*, *STAR*, *CYP11PA1* and *CYP19A1*), even from small antral to large antral follicle sizes [39]. In hindsight, a better comparison would have been with granulosa cells derived from primordial follicles, which we have never been successful at isolating.

The fetal ovarian fibroblasts were similar to adult fibroblasts in their gene expression profiles. However, in order to isolate them free from GREL cells we cultured them and compared them with adult fibroblasts which were also cultured. We have previously observed that ovarian fetal fibroblasts from fetuses in their first trimester of gestation express and secrete fibrillin 3 [8, 40], but they lose that ability rapidly in culture and instead express fibrillin 1 [26]. However, the cultured fetal fibroblasts did retain some fetal characteristics as TGFβ1 decreased expression of *COL1A1* and *COL3A1*, as happens with fetal fibroblasts and which is the opposite of what adult fibroblasts do [41, 42]. We also examined the effects of 24 hormones, stimulators and growth factors and analysed the expression of genes expressed in fibroblasts (*NR2F2*, *FBN1*, *COL1A1* and *COL3A1*). Only TGFβ1 altered the expression of three of these genes. We were also able to compare the expression patterns of genes in fibroblasts isolated in a previous experiment which were cultured with no added EGF [26] with those isolated in the current experiments isolated with EGF. The result showed that EGF significantly increased *FBN1* expression, but decreased the gene expression of *COL1A1*. Thus fetal fibroblasts appear to have limited number of regulators.

In this study *NR2F2*, *FBN1*, *COL1A1* and *COL1A3* were used as the marker genes for the stromal fibroblasts in the ovary [30, 43, 44]. *NR2F2* encodes COUP-TF2 and it is abundantly expressed in the human fetal ovary, playing a role as a sex-determining factor in the female gonad [45]. *FBN1*, *COL1A1*, *COL1A3* which encode fibrillin 1, alpha 1 chain of collagen type I and alpha 1 chain of collagen type , respectively, are three components in ECM. Fibrillin 1 is a glycoprotein essential for the formation of microfibrils, which appear to provide a scaffold for the tissues [46]. Moreover, fibrillin networks exhibit regulatory functions on cell signalling and behaviour [47]. Collagens interact with fibres in ECM and regulate the cell morphology, growth and migration [48, 49], and collagen type I and III are secreted by fibroblasts [48, 50]. The high expression of *NR2F2*, *FBN1* and *COL1A1* in fetal fibroblasts was consistent with the higher expression in adult fibroblasts than ovarian surface epithelial cells and granulosa cells. *LAMB2* encodes the beta 2 chain of laminin, which is a major component in basal laminas and essential to cell adhesion [51, 52]. Expression of *LAMB2* and *COL4A1* increased slightly across gestation suggesting that more basal lamina is present in late gestational fetal ovaries, which

was consistent with the formation and growth of follicles and maturation of the ovarian surface epithelium.

In genes involved in steroidogenesis, *NR5A1* encodes steroidogenic factor 1, and it could regulate genes encoding enzymes for steroidogenesis by binding to the promoter regions of genes encoding steroidogenesis enzymes, such as *CYP11A1*, *CYP19A1* and *HSD3B1* [53, 54]. *STAR*, also regulated by steroidogenic factor 1 [54], encodes steroidogenic acute regulatory protein, which is a transport protein for the transfer of cholesterol for the production of progesterone [55]. Our current data showed that *NR5A1* was predominantly expressed in adult epithelial cells and granulosa cells, and this was consistent with the higher *NR5A1* expression in GREL cells than fetal fibroblasts, suggesting that GREL cells exhibited a similar pattern with its adult counterparts in *NR5A1* expression. *CYP11A1*, *HSD3B1*, *CYP19A1* and *HSD17B1* encode different enzymes in steroidogenesis, and they were predominantly expressed in granulosa cells. However, they were only minimally expressed in GREL cells.

Previous studies have demonstrated that oestrogens promote proliferation of ovarian surface epithelium [56] and granulosa cells [57]. *ESR1* and *ESR2* encode oestrogen receptor α and β, respectively. Our current data demonstrated that *ESR1* was mostly expressed in the ovarian surface epithelial cells and *ESR2* was predominantly expressed in granulosa cells, consistent with previous studies [58, 59]. The very low levels of expression of *ESR1* and *ESR2* in fetal and adult fibroblasts suggested that oestrogens do not directly regulate fibroblasts in ovarian development or function. Moreover, the weak expression of *ESR1* and *ESR2* in GREL cells suggested that GREL cells in culture are not regulated by oestrogens.

TGFβ signalling regulates processes such as follicle growth, oocyte meiotic maturation and luteinisation in the ovary [60, 61]. *TGFBR1* and *TGFBR2* encode TGFβ receptors 1 and 2, and have different affinities to TGFβ1, TGFβ2 and TGFβ3 [62–64]. Unlike these receptors, the type 3 receptor, encoded by *TGFBR3*, does not have any kinase activity, and it functions as an accessory receptor stabilising the complexes between type 1 and 2 receptors and modulating the activity of TGFβ ligands [65, 66]. Compared with the fetal ovary, GREL cells and fetal fibroblasts in culture had higher expression levels of *TGFBR1*, however, the two fetal ovarian cells did not have higher expression of *TGFBR2* or *TGFBR3* than the fetal ovary. Moreover, the higher expression levels of *TGFBR1*, *TGFBR2* and *TGFBR3* in ovarian surface epithelial cells suggests that they could be more sensitive to ligands of the TGFβ family than granulosa cells. Similarly, this scenario also happens in other genes, such as *KRT19*, *FBN1* and *STAR*, which implies the difference of granulosa cell from ovarian surface epithelial cell and GREL cell. Oocyte-derived factors, GDF9 and BMP15, regulate proliferation and differentiation of granulosa cells [23]. Therefore, oocyte derived factors would be more necessary in the differentiation of GREL cell into granulosa cell.

*CCND2*, *CCNE1* and *CCNE2* encode cyclin D2, cyclin E1 and cyclin E2, respectively, and activate cyclin-dependent kinases, which regulate cell cycle G1/S transition [67, 68] and increase cell proliferation [69–71]. Our previous study demonstrated that the proliferating index was decreasing in both, the non-stromal component and the stroma of the fetal ovary, and the expression of *CCND2* was decreasing across gestation [31, 32]. Consistent with the previous studies, *CCNE1* and *CCNE2* in the fetal ovary were also negatively correlated with the gestational age.

Aldehyde dehydrogenase 1 family member A1 is encoded by *ALDH1A1* and functions as an enzyme in the biosynthesis of retinoic acid which regulates cell functions, such as self-renewal and differentiation. Furthermore, *ALDH1A1* regulates both normal stem cells and tumor-initiating cells, and it is often used as a marker for stem cells [72, 73]. Leucine-rich repeat-containing G-protein coupled receptor 5 (LGR5) is encoded by *LGR5* and it plays an important role in cell self-renewal [74, 75]. LGR5-positive stem cells have been identified in

the mouse ovarian surface epithelium [21]. *OCT4* is required for the survival of primordial germ cells [76], and its expression decreases at the onset of meiotic prophase I [77, 78]. Depending on the species, *VASA* and *DAZL* are essential for the survival or migration of primordial germ cell [79, 80]. The observed low expression of *ALDH1A1*, *LGR5*, *OCT4*, *DAZL* and *VASA* in GREL cells and fetal fibroblasts indicated that the two cell types probably do not have properties of stem/germ cells.

## Conclusions

The two major somatic cell types of the developing ovary, GREL cells and fetal fibroblasts, have different cellular origins and exhibited different morphologies in cell culture. They also have distinct gene expression profiles. They also differ from the cells they ultimately differentiate in to. GREL cells had more epithelial properties which were closer to ovarian surface epithelial cells and less so to granulosa cells. Considering the regulation of the oocyte on granulosa cells, perhaps oocyte secreted factors might be necessary to establish the granulosa cell lineage. Fetal fibroblasts differed minimally from adult fibroblasts apart from higher expression of cell cycle genes. TGFβ1 and EGF may remodel ECM by regulating gene expression in fetal fibroblasts. In summary, whilst we can conclude that there are at least two somatic cells in the ovary with very different properties that also distinguish them from their adult counterparts, the regulation of cell fate determination, particularly of the bi-potentiality of GREL cells, remains to be elucidated.

## Supporting information

**S1 Fig. Behaviour of bovine fetal ovarian cells cultured on collagen type I coated plates.** After 2 days in culture, sporadic GREL cell clusters (A) and fetal fibroblasts (B) could be observed. GREL clusters grew slowly and formed larger clusters (C) in the subsequent culture days. GREL cells appeared polygonal, and in some areas tightly packed, as previously observed [8]. (D) Fetal fibroblasts grew faster than GREL cells and showed the typical spindle shaped appearance as seen in adult fibroblasts. (E) Alongside wells with pure GREL cells and pure fibroblasts, wells with both cell types could be observed. In these GREL cells formed clearly distinguishable clusters surrounded by fibroblasts. A sterile blunt-ended glass pipette was used to remove the fibroblasts and continue the culture to obtain GREL cells (F). Gestational ages were: (A) 51 and (B-F) 110 days. Bars: A-B, D-F = 100 μm; C = 200 μm. (PDF)

**S2 Fig. Representative micrograph images of each isolate of GREL cells used for qRT-PCR.** GREL cell clusters were grown on collagen type I coated plates. All 7 GREL cell samples collected for qRT-PCR were from passage 0. Gestational ages were (A) 51, (B) 56, (C) 73, (D, E) 110, (F) 127 and (G) 177 days. Bars: (A-G) = 100 μm. (PDF)

**S3 Fig. Representative micrograph images of each isolate of ovarian fetal fibroblast cells used for qRT-PCR.** Fetal fibroblasts were grown on collagen type I coated plates and collected on the same day as the corresponding GREL cells for gene expression analysis. All 6 fibroblast samples were from passage 0. Gestational ages were (A) 51, (B) 56, (C) 73, (D, E) 110, (F) 127 and (G) 177 days. Bars: (A-F) = 100 μm. (PDF)

**S4 Fig. Normalised gene expression of genes in GREL cells and fetal fibroblasts.** Gene expression levels in cultured GREL (n = 7) and fetal fibroblasts (n = 6) were normalised to

PPIA and RPL32, and they are presented as scatter plots.
(PDF)

**S5 Fig. Normalised gene expression of stromal genes in fetal fibroblasts in the presence of different treatments.** Fibroblasts from bovine fetal ovary (n = 5) were cultured with different treatments for 18 h. Fold changes of gene expression to the control groups are presented as mean ± SEM (normalised to *PPIA* and *RPL32*). Significantly different results among groups were determined by one-way ANOVA with Tukey's post-hoc test.
(PDF)

**S6 Fig. mRNA expression levels of stromal genes in fetal fibroblasts cultured in presence or absence of EGF.** Columns in white colour and black colour show gene expression in fetal fibroblasts cultured in the absence of EGF (n = 11) in a previous study [26] and in the presence of EGF (n = 7) in the current study, respectively. Data of gene expression are presented as mean ± SEM (normalised to *PPIA* and *RPL32*). Significantly different results between groups were determined by Student's t-test. ** $P < 0.01$.
(PDF)

**S1 Table. Treatments of cultured bovine fetal ovarian fibroblasts.**
(PDF)

**S2 Table. List of genes and primers used for qRT-PCR.**
(PDF)

**S3 Table. Pearson correlation coefficients (R) of expression levels of 10 newly-examined genes in bovine fetal ovaries and gestational age (n = 27).**
(PDF)

## Acknowledgments

We thank Thomas Foods International (Murray Bridge, SA), Strath Meats (Strathalbyn, SA) and The Midfield Group (Warrnambool, VIC) for providing bovine tissues and Mrs. Wendy M. Bonner for the collection of bovine fetal and adult ovaries.

## Author Contributions

**Conceptualization:** Menghe Liu, Katja Hummitzsch, Raymond J. Rodgers.

**Formal analysis:** Menghe Liu, Katja Hummitzsch, Raymond J. Rodgers.

**Funding acquisition:** Katja Hummitzsch, Richard A. Anderson, Raymond J. Rodgers.

**Investigation:** Menghe Liu, Katja Hummitzsch.

**Methodology:** Menghe Liu, Katja Hummitzsch, Nicole A. Bastian, Monica D. Hartanti, Qianhui Wan, Raymond J. Rodgers.

**Project administration:** Katja Hummitzsch, Raymond J. Rodgers.

**Resources:** Katja Hummitzsch, Raymond J. Rodgers.

**Supervision:** Katja Hummitzsch, Helen F. Irving-Rodgers.

**Writing – original draft:** Menghe Liu.

**Writing – review & editing:** Katja Hummitzsch, Richard A. Anderson, Raymond J. Rodgers.

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
