## [Decision Letter · Decision Letter 0]

8 Feb 2022

PONE-D-21-29973Characterisation of bovine ovarian fetal fibroblasts and gonadal ridge epithelial-like cells and comparison to their adult counterpartsPLOS ONE

Dear Dr. Rodgers,

Thank you for submitting your manuscript to PLOS ONE. After careful consideration, we feel that it has merit but does not fully meet PLOS ONE’s publication criteria as it currently stands. Therefore, we invite you to submit a revised version of the manuscript that addresses the points raised during the review process.

We look forward to receiving your revised manuscript.

Kind regards,

Academic Editor

PLOS ONE

Journal Requirements:

2. Thank you for stating the following in the Acknowledgments/ Funding Section of your manuscript: 

Funding was received from Adelaide University China Fee Scholarships (ML and QW; https://www.csc.edu.cn/chuguo), Adelaide Graduate Research Scholarship (NAB; http:// www.adelaide.edu.au/), Australia Awards Scholarship (MDH; https://www.dfat.gov.au/people-to-people/australia-awards/australia-awards-scholarships), Robinson Research Institute Career Development Fellowship (KH; https://www.adelaide.edu.au/robinson-research-institute/) and Building On Ideas Grant (KH; https://www.adelaide.edu.au/robinson-research-institute/), National Health and Medical Research Council of Australia Centre for Research Excellence in the Evaluation, Management and Health Care Needs of Polycystic Ovary Syndrome (RJR; GTN1078444; https://www. nhmrc.gov.au/) and The Centre for Research Excellence on Women’s Health in Reproductive life (RJR and KH; GTN1171592; https://www. nhmrc.gov.au/) and the UK Medical Research Council (RAA; grant No. G1100357; https://mrc.ukri.org/). The funders did not play any role in the study design, data collection and analysis, decision to publish, or preparation of the manuscript.

Funding was received from Adelaide University China Fee Scholarships (ML and QW; https://www.csc.edu.cn/chuguo), Adelaide Graduate Research Scholarship (NAB; http:// www.adelaide.edu.au/), Australia Awards Scholarship (MDH; https://www.dfat.gov.au/people-to-people/australia-awards/australia-awards-scholarships), Robinson Research Institute Career Development Fellowship (KH; https://www.adelaide.edu.au/robinson-research-institute/) and Building On Ideas Grant (KH; https://www.adelaide.edu.au/robinson-research-institute/), National Health and Medical Research Council of Australia Centre for Research Excellence in the Evaluation, Management and Health Care Needs of Polycystic Ovary Syndrome (RJR; GTN1078444; https://www. nhmrc.gov.au/) and The Centre for Research Excellence on Women’s Health in Reproductive life (RJR and KH; GTN1171592; https://www. nhmrc.gov.au/) and the UK Medical Research Council (RAA; grant No. G1100357; https://mrc.ukri.org/). The funders did not play any role in the study design, data collection and analysis, decision to publish, or preparation of the manuscript.

RAA reports consultancy work for Ferring, Merck, IBSA, Roche Diagnostics, NeRRe Therapeutics and Sojournix Inc. The other authors of this manuscript have nothing to declare and no conflict of interest that could be perceived as prejudicing the impartiality of the research reported.

Reviewers' comments:

Reviewer's Responses to Questions

**Comments to the Author**

1. Is the manuscript technically sound, and do the data support the conclusions?

Reviewer #1: Yes

Reviewer #2: Yes

Reviewer #3: Yes

Reviewer #4: Partly

2. Has the statistical analysis been performed appropriately and rigorously? 

Reviewer #1: Yes

Reviewer #2: Yes

Reviewer #3: Yes

Reviewer #4: Yes

3. Have the authors made all data underlying the findings in their manuscript fully available?

Reviewer #1: Yes

Reviewer #2: Yes

Reviewer #3: Yes

Reviewer #4: No

4. Is the manuscript presented in an intelligible fashion and written in standard English?

Reviewer #1: Yes

Reviewer #2: Yes

Reviewer #3: Yes

Reviewer #4: Yes

5. Review Comments to the Author

Reviewer #1: In this manuscript “Characterisation of bovine ovarian fetal fibroblasts and gonadal ridge epithelial-like cells and comparison to their adult counterparts' ' reports importance of cell types in ovary development. This study can be helpful to have more knowledge on ovary development and ovary diseases. Article has details of methods and results, which are presented well in the manuscript. This article has made careful and further analysis of ovary development from the previous Plos One articles from these authors. This topic seems to help readers connect with the author's previous study in the features cell types in ovary development. I would recommend publication of this study.

Minor comment:

In line 496 authors should correct “week” to weak.

Authors can add a flowchart how to collect and maintain fetal fibroblasts and GREL1 cells from ovary tissues.

Reviewer #2: Reviewer’s comment

In the study titled ‘Characterization of bovine ovarian fetal fibroblasts and gonadal ridge epithelial-like cells and comparison to their adult counterparts’ the authors Liu et al isolated fibroblasts and gonadal ridge epithelial-like cells (GREL) from fetal ovary of pregnant cow. They have also isolated fibroblasts, surface epithelial cells, and granulosa cells from adult ovaries from non-pregnant cows. The culture of cells was done without passage, and based on the morphology, the contaminant cell if any, were removed by scraping. The cells were harvested and frozen for gene expression studies. To identify possible regulators of extracellular matrix, the fetal ovary fibroblasts were treated with hormones, stimulators, and growth factors. The expression profile of various genes was studied.

The two somatic cells, GREL cells and fetal fibroblasts, were distinguishable in terms of cellular origins, morphology, and gene expression profile in cell culture. These cells are also distinguishable from their adult counterparts. TGFβ1 and EGF may remodel ECM by regulating gene expression in fetal fibroblasts.

The article is a preliminary work in understanding the development of the ovary. The authors have put a good effort in preparing the manuscript. However, for the readers, it may a bit complicated. Hence need to be simplified. The following suggestions might increase the value of the manuscript.

1. The authors may consider changing the title to have a clear differentiation from their previous publication (Hummitzsch et al. 2019. Morphometric analyses and gene expression related to germ cells, GREL cells, and granulosa cells during development of the bovine fetal ovary). The suggested new title is ‘Isolation, culture, and characterization of bovine fetal ovarian fibroblasts and gonadal ridge epithelial-like cells compared to their adult counterparts.

2. The authors may consider simplifying the abstract. As in the discussion, group names of the genes may be mentioned.

3. Line 53: In culture, the cell types were morphologically different. - Rephrase the sentence. ‘In culture, these two cell types were morphologically different.

4. Published studies from the literature on isolating the ovarian cells are missing in the introduction. Studies in other species also can be included.

5. The methodology can be explained with reasons such that readers will understand the purpose.

6. The basis of selecting the genes can be explained in methods.

7. More details on samples collection can be included in methods (how many days of pregnancy, age of cow etc), although some of them have been explained in figure /table legends.

8. Line 126-137 After digesting with collagenase, how do you remove the undigested cell debris? A low-speed spin first or filtration?

9. PCR is a sensitive technique. What are the precautions taken to prevent fetal ovary fibroblasts and GREL cells from getting contaminated with other cells?

10. Any specific advantage of washing the tissues with 70% alcohol over washing it with media with antibiotics and antifungals?

11. Fig S4: X-axes title to be included.

12. Fig S5: Is there any statistically significant expression? If yes, the significance can be indicated by asterisks* as in Fig S6.

13. Figure S1 can be moved to the main body, or a representative figure of various cells may be included.

Reviewer #3: Characterisation of bovine ovarian fetal fibroblasts and gonadal ridge epithelial like cells and comparison to their adult counterparts

Menghe Liu et al

This paper seeks to isolate and characterise fibroblasts and GREL cells, the 2 major components of the fetal ovarian stroma. In the context of ovarian development, the developmental origins of health and disease (notably PCOS) the work is particularly relevant and important and, perhaps understated somewhat in the introduction.

The work is of a good standard, the results are of interest and build on our knowledge in this area. The manuscript is in general well crafted and put together. There are occasional lapses in grammar but these do not impact on the message.

Introduction

Lines 84-88. Give the discussion later around GREL cell differentiation into surface epithelial cells and granulosa cells I believe some mention or reference to the oocytes impacts of granulosa cell function/development would be appropriate (e.g. McNatty et al, Reproduction vol 128, 2004).

Minor issues

Lines94-95. Grammar could be addressed. Remove into or replace with thereby. ?

Line 105. A little understated I would have thought

Materials and methods. What I have struggled to find is the number of fetuses that were collected for the isolation of fibroblasts and GREL cells and importantly the age range of these fetuses. This should be clearly laid out in the M&M somewhere or in the figure legends.

Line 141 Grammar could be addressed here.

Lines 157-171. Isolation of adult ovarian fibroblasts. It is unclear to me and would be interesting to know whether the tissue to be minced and subsequently cultured contained any follicles

Line 254-255. While some of this data has been previously published it is my opinion that inclusion of this data is warranted

Presentation of gene expression data. Can any inference be drawn from the differences in expression levels between fetal ovaries and the isolated fetal cells, could this be an “artifact” of culture?. It should be acknowledged that for the adult cells fibroblasts have been cultured whereas both granulosa cells and OSE have not. Therefore differences between fibroblasts and granulosa or OSE may be attributed the culture of the cells. The same could be argued for differences between GREL cells and granulosa or OSE. If the authors do not accept this then their reason should be outlined in the discussion.

Line 268 and 272. it is OCLN not OCLD?

Lines 301-302. At the low levels of expression seen is the data reliable enough to support a significant difference, has the PCR been validated at these levels of expression.

Lines 407-408.Would the authors care to speculate of the difference in FOXL2 expression between this work and the previous IHC work. Could this be a function of culturing cells?.

Lines 421-422 and 438-440. This would not be surprising given that the granulosa cells have been under the influence of oocytes for some time.

Line 478. This would be consistent with the formation and growth of follicles

Line 497. The caveat with this statement ( and a number of others in the discussion) is that we are talking about expression in cultured cells. This should really be mentioned in the discussion.

Lines 503-505. This scenario is true for other genes as well (eg FBN1, STAR and KRT19). What is the implication of this pattern of expression.

Lines 533-534. I expected to see more around this statement in the discussion.

Figures and Legends. These are acceptable, well formatted and convey the intended message well. The exception is Supplementary Fig 1C which has poor uneven lighting and should be replaced

Discussion

Line 279. GJA1 was higher in granulosa cells not less.

Line 330-331. It would appear that for CCNE1 and CCNE2 the pattern of expression over gestation does not necessarily follow a linear pattern. The possibility should be acknowledged in the results or addressed with a more complex analysis.

Line 335. The difference in CCNE2 expression between adult OSE and adult fibroblasts is not indicated in the actual figure.

Reviewer #4: The manuscript "Characterisation of bovine ovarian fetal fibroblasts and gonadal ridge epithelial-like

cells and comparison to their adult counterparts" is basically a study where differential genes that are expressed in bovine fetal and adult tissue are compared after specific cell types have been cultured to isolate them.

General comments- A major problem with the manuscript is that it describes lists of genes different but does not explain why the differences in genes between each isolated cell population is important or what the different genes might confer to their functionality. Also if these genes are going to be utilized as markers of specific genes at specific ages (which would be a good enough function) then the processes they are involved in should be better described and there also needs to be better functional data to support these claims. Also the differences in the genes may also demonstrate why these cells are morphologically different and then how that helps their function.

As is-- the manuscript leaves the reviewer questioning what each gene is: As there are no genes that are defined and then what the function of the gene is--and how the group of genes that are differentially expressed--what does that mean? The authors appear to be demonstrating how different and similar fetal tissues are to adult but they need to develop a graphical abstract or diagram to demonstrate how these different fetal tissues may ultimately differentiate into adult--or what genes demonstrate different cell lineages (what genes are markers or are enriched for specific genes and that makes them have a specific lineage or type of cell). However, no identification of processes or function is made and the reader is left to see what genes are different but not really have a good idea of why these differences may be important.

Also no functional data other that treatment of TGFbeta and EGF is conducted. The outcome of these treatments are not fully incorporated into what the relevance of the differential gene expression observed means. Aspects of the paper are interesting but it does remain a characterization study throughout the paper and lacks some functional aspects that are necessary to give the paper more significance to its findings.

6. PLOS authors have the option to publish the peer review history of their article (what does this mean?). If published, this will include your full peer review and any attached files.

Reviewer #1: No

Reviewer #2: No

Reviewer #3: No

Reviewer #4: No

---

## [Author Response · Author response to Decision Letter 0]

22 Mar 2022

Note the line numbers refer to the track changes version of the manuscript.

Reviewer #1: 

COMMENT: In line 496 authors should correct “week” to weak.

REPLY: Done.

COMMENT: Authors can add a flowchart how to collect and maintain fetal fibroblasts and GREL1 cells from ovary tissues.

REPLY: We prefer not to do this and keep the methods presented simply. 

Reviewer #2: 

COMMENT: 1. The authors may consider changing the title to have a clear differentiation from their previous publication (Hummitzsch et al. 2019. Morphometric analyses and gene expression related to germ cells, GREL cells, and granulosa cells during development of the bovine fetal ovary). The suggested new title is ‘Isolation, culture, and characterization of bovine fetal ovarian fibroblasts and gonadal ridge epithelial-like cells compared to their adult counterparts.

REPLY: Done.

COMMENT: 2. The authors may consider simplifying the abstract. As in the discussion, group names of the genes may be mentioned.

REPLY: We think listing gene names in the abstract would be helpful to the reader. No changes were made.

COMMENT: 3. Line 53: In culture, the cell types were morphologically different. - Rephrase the sentence. ‘In culture, these two cell types were morphologically different.

REPLY: Done (Line 52).

COMMENT: 4. Published studies from the literature on isolating the ovarian cells are missing in the introduction. Studies in other species also can be included.

REPLY: Whilst this is an important point it is something that could be done later in a methods review. To do this would potentially detract from the lines of reasoning that we have in our introduction.

COMMENT: 5. The methodology can be explained with reasons such that readers will understand the purpose.

REPLY: Done. We now say ‘We had to adopt this approach because we had not identified a specific antigen on surface of GREL cells to allow us to utilise more direct cell isolation methods such as the use of magnetic beads or fluorescence-activated cell sorting.’ (Line 399-402)

COMMENT: 6. The basis of selecting the genes can be explained in methods.

REPLY: Done. We now say “We analysed the expression levels of 34 genes relating to cell junctions, extracellular matrix, enzymes and transcription factors in order to characterise the cultured GREL cells and fetal fibroblasts” (Line 231-233).

COMMENT: 7. More details on samples collection can be included in methods (how many days of pregnancy, age of cow etc), although some of them have been explained in figure /table legends.

REPLY: We have now added the days of pregnancy into the relevant figure legends. (Pages 26-28) 

COMMENT: 8. Line 126-137 After digesting with collagenase, how do you remove the undigested cell debris? A low-speed spin first or filtration?

REPLY: We now say “The cell suspensions were then passed through cell strainers and centrifuged at 1200 rpm for 5 min and the supernatants were discarded” (Line 142-143).

COMMENT: 9. PCR is a sensitive technique. What are the precautions taken to prevent fetal ovary fibroblasts and GREL cells from getting contaminated with other cells?

REPLY: When cDNA was synthesised, RNA was isolated from only one cell type at one time. When cDNA samples were added into the PCR systems, cDNA samples from different cell types was not added at one time.

COMMENT: 10. Any specific advantage of washing the tissues with 70% alcohol over washing it with media with antibiotics and antifungals?

REPLY: Since our samples were collected from the local abattoir, washing the tissue with 70% alcohol has a greater ability to reduce contamination in the subsequent cell culture. 

COMMENT: 11. Fig S4: X-axes title to be included.

REPLY: The X-axes title has been added in Fig S4.

COMMENT: 12. Fig S5: Is there any statistically significant expression? If yes, the significance can be indicated by asterisks* as in Fig S6.

REPLY: There is no statistical significant differences, and this has been described in the results (Line 384).

COMMENT: 13. Figure S1 can be moved to the main body, or a representative figure of various cells may be included.

REPLY: We now include a representative figure (Fig 1)

Reviewer #3: 

COMMENT: 1. Lines 84-88. Give the discussion later around GREL cell differentiation into surface epithelial cells and granulosa cells I believe some mention or reference to the oocytes impacts of granulosa cell function/development would be appropriate (e.g. McNatty et al, Reproduction vol 128, 2004).

REPLY: Now we say “The differences between the two cell types which GREL cells differentiate into could be the result of regulation by oocyte-secreted factors such as growth differentiation factor 9 (GDF9) and bone morphogenetic protein 15 (BMP15) which play essential roles in regulating the proliferation and differentiation of granulosa cells [23]. The ovarian surface epithelium, on the other hand, is separated from oocytes by stroma and basal lamina and is likely not exposed to oocyte-secreted factors” in Line 93-100.

COMMENT: 2. Lines 94-95. Grammar could be addressed. Remove into or replace with thereby. ?

REPLY: Done (Line 103-104).

COMMENT: 3. Line 105. A little understated I would have thought

REPLY: We now say ‘To conduct the first examination of GREL cells and to compare them to fetal fibroblasts,…..’ (Line 114-115)

COMMENT: 4. Materials and methods. What I have struggled to find is the number of fetuses that were collected for the isolation of fibroblasts and GREL cells and importantly the age range of these fetuses. This should be clearly laid out in the M&M somewhere or in the figure legends.

REPLY: The ages of fetuses that were collected for fibroblasts and GREL cells have been added in the figure legends. Pages 26, 27 and 28.

COMMENT: 5. Line 141 Grammar could be addressed here.

REPLY: Done. We now say “Then 250 cells were seeded per well in 24-well plates which were …” (Line 151-152).

COMMENT: 6. Lines 157-171. Isolation of adult ovarian fibroblasts. It is unclear to me and would be interesting to know whether the tissue to be minced and subsequently cultured contained any follicles.

REPLY: We have added some additional information to indicate that we avoided areas with follicles. We now say ‘Individual ovaries were cut into quarters and tissue areas devoid of visible follicles was cut to a depth of about 0.5 mm from the surface and minced with sterile razor blades…’

COMMENT: 7. Line 254-255. While some of this data has been previously published it is my opinion that inclusion of this data is warranted

REPLY: The data are shown in the figs. Its just the statement on line 268-269 that gave the wrong message and we have amended it. 

COMMENT: 8. Presentation of gene expression data. Can any inference be drawn from the differences in expression levels between fetal ovaries and the isolated fetal cells, could this be an “artifact” of culture? It should be acknowledged that for the adult cells fibroblasts have been cultured whereas both granulosa cells and OSE have not. Therefore differences between fibroblasts and granulosa or OSE may be attributed the culture of the cells. The same could be argued for differences between GREL cells and granulosa or OSE. If the authors do not accept this then their reason should be outlined in the discussion.

REPLY: Yes we accept this comment. We now say in the opening paragraph of the Discussion ‘In this study we developed methods for isolating and culturing fetal ovarian GREL cells and fetal ovarian fibroblasts by removal of contaminating cells during culture and expansion from single colonies without passage. We had to adopt this approach because we had not identified a specific antigen on the surface of GREL cells to allow us to utilise more direct cell isolation methods such as the use of magnetic beads or fluorescence-activated cell sorting. We investigated the expression of a variety of genes to identify properties of GREL cells and fetal fibroblasts and examined differences with their adult counterparts. It should be acknowledged that culturing of GREL cells could have contributed to differences observed with the granulosa cells or ovarian surface epithelial cells which were not cultured’.

COMMENT: 8. Line 268 and 272. it is OCLN not OCLD?

REPLY: This error has been corrected throughout the manuscript.

COMMENT: 9. Lines 301-302. At the low levels of expression seen is the data reliable enough to support a significant difference, has the PCR been validated at these levels of expression.

REPLY: The Ct values of CYP11A1 and INHBA of the fetal fibroblast samples were all below 30. However, in the PCR examination of the blank controls of the two genes, Ct values were above 35 or no Ct values were obtained. Therefore, we believed that the two genes were expressed in fetal fibroblasts, even if they were in very low levels.

COMMENT: 10. Lines 407-408.Would the authors care to speculate of the difference in FOXL2 expression between this work and the previous IHC work. Could this be a function of culturing cells?.

REPLY: We are not sure why this is the case at this stage.

COMMENT: 11. Lines 421-422 and 438-440. This would not be surprising given that the granulosa cells have been under the influence of oocytes for some time.

REPLY: We agree with this opinion. The addition of sentences to the introduction address this issue directly (Line 93-100).

COMMENT: 12. Line 478. This would be consistent with the formation and growth of follicles

REPLY: We agree with this opinion, and this statement has been added in Line 501-502.

COMMENT: 13. Line 497. The caveat with this statement ( and a number of others in the discussion) is that we are talking about expression in cultured cells. This should really be mentioned in the discussion.

REPLY: This has been mentioned in Line 409, 431, 432, 521 and 527.

COMMENT: 14. Lines 503-505. This scenario is true for other genes as well (eg FBN1, STAR and KRT19). What is the implication of this pattern of expression.

REPLY: We think that the big differences between granulosa cell and other cell types (GREL cell, ovarian surface epithelial cell and fibroblast) imply the oocyte-derived regulation affects the differentiation from GREL cell into granulosa cell. Another statement has been added in the discussion in Line 531-535.

COMMENT: 15. Lines 533-534. I expected to see more around this statement in the discussion.

REPLY: We have added more statement in Line 531-535.

COMMENT: 16. Figures and Legends. These are acceptable, well formatted and convey the intended message well. The exception is Supplementary Fig 1C which has poor uneven lighting and should be replaced.

REPLY: Done.

COMMENT: 17. Line 279. GJA1 was higher in granulosa cells not less.

REPLY: This error has been corrected in Line 294.

COMMENT: 18. Line 330-331. It would appear that for CCNE1 and CCNE2 the pattern of expression over gestation does not necessarily follow a linear pattern. The possibility should be acknowledged in the results or addressed with a more complex analysis.

REPLY: We chose linear models so that we only identified strong relationships. There were many identified. No changes made.

COMMENT: 19. Line 335. The difference in CCNE2 expression between adult OSE and adult fibroblasts is not indicated in the actual figure.

REPLY: There is no statistically significant difference between adult ovarian surface epithelial cells and adult fibroblasts. We now say ‘….however, there was no statistically significant difference between adult ovarian surface epithelial cells and adult fibroblasts.’ (Line 353-354).

Reviewer #4: 

COMMENT A major problem with the manuscript is that it describes lists of genes but does not explain why the differences in genes between each isolated cell population is important or what the different genes might confer to their functionality. Also if these genes are going to be utilized as markers of specific genes at specific ages (which would be a good enough function) then the processes they are involved in should be better described and there also needs to be better functional data to support these claims. Also the differences in the genes may also demonstrate why these cells are morphologically different and then how that helps their function.

As is the manuscript leaves the reviewer questioning what each gene is: As there are no genes that are defined and then what the function of the gene is--and how the group of genes that are differentially expressed--what does that mean? The authors appear to be demonstrating how different and similar fetal tissues are to adult but they need to develop a graphical abstract or diagram to demonstrate how these different fetal tissues may ultimately differentiate into adult--or what genes demonstrate different cell lineages (what genes are markers or are enriched for specific genes and that makes them have a specific lineage or type of cell). However, no identification of processes or function is made and the reader is left to see what genes are different but not really have a good idea of why these differences may be important.

REPLY: The selection of the genes emanates from our previous publication (Hummitzsch et al, 2013) where we first describe GREL cells, their morphology and some microarray data. We also chose genes known to be expressed in fibroblasts. In this study, we developed a method to isolate and culture GREL cells, and the result showed that these GREL cells were mostly consistent with our previous study. The fibroblasts expressed genes as expected. This this validates our cell isolation and then we compared the cells with themselves and with adult counterparts. Thus providing new knowledge.

COMMENT: Also no functional data other that treatment of TGFbeta and EGF is conducted. The outcome of these treatments are not fully incorporated into what the relevance of the differential gene expression observed means. Aspects of the paper are interesting but it does remain a characterization study throughout the paper and lacks some functional aspects that are necessary to give the paper more significance to its findings.

REPLY: We do not disagree that the study is mostly a characterization study. However, this is the first time to show that GREL cells are a real entity. They are very different somatic cell to fibroblasts and they are not granulosa cells nor surface epithelial cells. It is the first time they have been isolated and studied.

---

## [Decision Letter · Decision Letter 1]

12 Apr 2022

PONE-D-21-29973R1Isolation, culture, and characterisation of bovine ovarian fetal fibroblasts and gonadal ridge epithelial-like cells and comparison to their adult counterpartsPLOS ONE

Dear Dr. Rodgers,

Thank you for submitting your manuscript to PLOS ONE. After careful consideration, we feel that it has merit but does not fully meet PLOS ONE’s publication criteria as it currently stands. Therefore, we invite you to submit a revised version (Minor Revision) of the manuscript that addresses the points raised during the review process.

We look forward to receiving your revised manuscript.

Kind regards,

Academic Editor

PLOS ONE

Journal Requirements:

Reviewers' comments:

Reviewer's Responses to Questions

**Comments to the Author**

1. If the authors have adequately addressed your comments raised in a previous round of review and you feel that this manuscript is now acceptable for publication, you may indicate that here to bypass the “Comments to the Author” section, enter your conflict of interest statement in the “Confidential to Editor” section, and submit your "Accept" recommendation.

Reviewer #2: All comments have been addressed

Reviewer #3: All comments have been addressed

Reviewer #4: All comments have been addressed

Reviewer #5: (No Response)

2. Is the manuscript technically sound, and do the data support the conclusions?

Reviewer #2: Yes

Reviewer #3: Yes

Reviewer #4: Yes

Reviewer #5: Partly

3. Has the statistical analysis been performed appropriately and rigorously? 

Reviewer #2: Yes

Reviewer #3: Yes

Reviewer #4: Yes

Reviewer #5: Yes

4. Have the authors made all data underlying the findings in their manuscript fully available?

Reviewer #2: Yes

Reviewer #3: Yes

Reviewer #4: Yes

Reviewer #5: Yes

5. Is the manuscript presented in an intelligible fashion and written in standard English?

Reviewer #2: Yes

Reviewer #3: Yes

Reviewer #4: Yes

Reviewer #5: No

6. Review Comments to the Author

Reviewer #2: The author’s response and changes made in the manuscript are satisfactory. The manuscript is improved now.

Reviewer #3: I am comfortable that the questions raised have been addressed to a satisfactory level. While I agree with reviewer 4 that more functional studies would have been useful i believe there is enough new and interesting material to warrant publication as a largely characterization study.

Reviewer #4: All concerns by this reviewer were met by authors. While a graphical abstract would help tell the story this reviewer is not going to make it a requirement for publication.

Reviewer #5: Isolation, culture, and Characterization of bovine ovarian fetal fibroblasts and gonadal ridge epithelial-like cells and comparison to their counterparts

Main Theme of the manuscript:

In this article, the authors have isolated gonadal ridge epithelial-like cells (GREL) and examined the expression of various genes in GREL cells along with other counterpart cells such as bovine ovarian fetal fibroblasts, adult fibroblasts, granulosa cells (GC), and ovarian surface cells (OSE) by qRT-PCR. The gene expression analysis shows that GREL cells have a differential pattern of gene expression compared with other cells. I appreciate the author's attempts at the basic characterization of GREL cells for the first time. However, this article is lacking some details as mentioned below.

Major comments:

This manuscript is mainly focused on just technical details of isolation, culture, and gene expression analysis. The authors listed out the differential expression of a list of genes but didn’t explain what it could be the effect of these differential gene expressions make them unique in terms of functionality of the GREL cells. To justify the title of the manuscript, authors must fully characterize the cells by performing an analysis of cell lineage markers, cell doubling time, cell cycle analysis, etc., and compare them with other counterpart cells. This would demonstrate why these cells are morphologically and functionally different from other cells. Since it is just isolation, culture, and gene expression analysis, I feel it could be a short communication or a technical report instead of a full-length research article.

Minor comments:

1. There are a few minor typos and grammatical errors throughout. The author must check the grammatical errors thoroughly.

2. It would give a good look at the gene expression data if it was shown in a heat-map instead of bar charts. Just one glance would give a complete picture of up and down-regulated genes across the various cell types. For example, the heat-map shows down-regulated genes in green and up-regulated genes in red in the heat-map.

3. What is the reason for adding 3 ng/ml EGF while isolating GREL and fibroblast cells?

4. Why is the incubation temperature at 38.5 c? is there any specific reason to keep them at that temperature?

5. How pure these cells are? how did the authors make sure that they didn’t pick up other cells while scrapping off the colonies? since PCR is a very sensitive technique, authors may get the expression of the genes from the cross-contaminated cells as well. Did the authors include fibroblast lineage marker genes as a control to check the purity of the GREL cells in the PCR panel?

6. Did the authors check the genomic DNA contamination by including non-RT controls? If yes it must be included in the methods section.

7. It is interesting to see the effect of TGF-b, and EGF treatment on gene expression but the authors didn’t explain the functional outcome. The manuscript lacks explanations for functional aspects although they are significant findings.

8. X and Y axis titles must be present for every graph.

9. In the figure, S4 scatter plot labels are missing, what are the blue and orange dots represent? What does n mean? the number of colonies or number of samples? please mention it in the figure legend.

7. PLOS authors have the option to publish the peer review history of their article (what does this mean?). If published, this will include your full peer review and any attached files.

Reviewer #2: No

Reviewer #3: No

Reviewer #4: No

Reviewer #5: **Yes: **Vijaya kumar Pidugu

---

## [Author Response · Author response to Decision Letter 1]

20 Apr 2022

COMMENT: This manuscript is mainly focused on just technical details of isolation, culture, and gene expression analysis. The authors listed out the differential expression of a list of genes but didn’t explain what it could be the effect of these differential gene expressions make them unique in terms of functionality of the GREL cells. To justify the title of the manuscript, authors must fully characterize the cells by performing an analysis of cell lineage markers, cell doubling time, cell cycle analysis, etc., and compare them with other counterpart cells. This would demonstrate why these cells are morphologically and functionally different from other cells. Since it is just isolation, culture, and gene expression analysis, I feel it could be a short communication or a technical report instead of a full-length research article.

REPLY: It should be appreciated that this is the first time GREL cells have been isolated and cultured and as such this a substantial contribution to field of fetal ovary development. We examined expression of 34 genes and compared those with their counterparts. This is a necessary first step in understanding these cells. Hence at this time there are no lineage markers discovered much less are they available to be monitored. 

If you feel it can be a short communication then we think the editor should decide this. 

COMMENT 1. There are a few minor typos and grammatical errors throughout. The author must check the grammatical errors thoroughly.

REPLY: Thank you for the suggestion. We have proofed it again. 

COMMENT 2. It would give a good look at the gene expression data if it was shown in a heat-map instead of bar charts. Just one glance would give a complete picture of up and down-regulated genes across the various cell types. For example, the heat-map shows down-regulated genes in green and up-regulated genes in red in the heat-map.

REPLY: Yes the heat maps could be useful, but we went to a lot of work to ensure we also had data on expression during the continuum of gestation and those data do not lend themselves readily to a heat map, hence we did not include one. 

COMMENT 3. What is the reason for adding 3 ng/ml EGF while isolating GREL and fibroblast cells?

REPLY: We found that GREL cells, being an epithelial cell, needed EGF to maintain the morphological phenotype. 

COMMENT 4. Why is the incubation temperature at 38.5 c? is there any specific reason to keep them at that temperature?

REPLY: The bovine body temperature is 38.5 C.

COMMENT 5. How pure these cells are? how did the authors make sure that they didn’t pick up other cells while scrapping off the colonies? since PCR is a very sensitive technique, authors may get the expression of the genes from the cross-contaminated cells as well. Did the authors include fibroblast lineage marker genes as a control to check the purity of the GREL cells in the PCR panel?

REPLY: Yes its possible that any contamination could lead to a PCR signal. However, we were using quantitative PCR and not using it for detection. So the comment about the sensitivity of PCR is not so relevant in this situation. Also Figs S2 and S3 show photos of each preparation of cells that were actually used in these experiments and clearly they are highly enriched for their respective cell types. 

In any event we said ‘In this study NR2F2, FBN1, COL1A1 and COL1A3 were used as the marker genes for the stromal fibroblasts in the ovary’ and these were higher in fibroblasts than GREL cells. We are confident of our results. 

COMMENT 6. Did the authors check the genomic DNA contamination by including non-RT controls? If yes it must be included in the methods section.

REPLY: Yes. We assumed this was a routine control that everyone did. In any event as we reported we used primers spanning introns. We now include this information in the methods (line 207).

COMMENT 7. It is interesting to see the effect of TGF-b, and EGF treatment on gene expression but the authors didn’t explain the functional outcome. The manuscript lacks explanations for functional aspects although they are significant findings.

REPLY: We said ‘TGFβ1 and EGF may remodel ECM by regulating gene expression in fetal fibroblasts.’ Given that we do not know about the in vivo expression of these molecules we feel in premature to speculate too much at this stage about the meaning of the results for the in vivo situation. 

COMMENT 8. X and Y axis titles must be present for every graph.

REPLY: We prefer to have the X-axis labelled for the Figs as a whole not as individual parts of the Fig.

9. In the figure, S4 scatter plot labels are missing, what are the blue and orange dots represent? What does n mean? the number of colonies or number of samples? please mention it in the figure legend.

REPLY: These are now clarified in the legend.

---

## [Decision Letter · Decision Letter 2]

1 May 2022

Isolation, culture, and characterisation of bovine ovarian fetal fibroblasts and gonadal ridge epithelial-like cells and comparison to their adult counterparts

PONE-D-21-29973R2

Dear Dr. Rodgers,

We’re pleased to inform you that your manuscript has been judged scientifically suitable for publication and will be formally accepted for publication once it meets all outstanding technical requirements.

Kind regards,

Academic Editor

PLOS ONE

Additional Editor Comments (optional):

Reviewers' comments:

Reviewer's Responses to Questions

**Comments to the Author**

1. If the authors have adequately addressed your comments raised in a previous round of review and you feel that this manuscript is now acceptable for publication, you may indicate that here to bypass the “Comments to the Author” section, enter your conflict of interest statement in the “Confidential to Editor” section, and submit your "Accept" recommendation.

Reviewer #2: All comments have been addressed

Reviewer #4: All comments have been addressed

Reviewer #5: All comments have been addressed

2. Is the manuscript technically sound, and do the data support the conclusions?

Reviewer #2: Yes

Reviewer #4: Yes

Reviewer #5: Yes

3. Has the statistical analysis been performed appropriately and rigorously? 

Reviewer #2: Yes

Reviewer #4: Yes

Reviewer #5: Yes

4. Have the authors made all data underlying the findings in their manuscript fully available?

Reviewer #2: Yes

Reviewer #4: Yes

Reviewer #5: Yes

5. Is the manuscript presented in an intelligible fashion and written in standard English?

Reviewer #2: Yes

Reviewer #4: Yes

Reviewer #5: Yes

6. Review Comments to the Author

Reviewer #2: Based on the reviewers comments the authors have made changes to the manuscript. Now the manuscript looks good.

The manuscript can be considered for publication.

Reviewer #4: (No Response)

Reviewer #5: (No Response)

7. PLOS authors have the option to publish the peer review history of their article (what does this mean?). If published, this will include your full peer review and any attached files.

Reviewer #2: No

Reviewer #4: No

Reviewer #5: No

---

## [Editor Report · Acceptance letter]

29 Jun 2022

PONE-D-21-29973R2 

Isolation, culture, and characterisation of bovine ovarian fetal fibroblasts and gonadal ridge epithelial-like cells and comparison to their adult counterparts 

Dear Dr. Rodgers:

I'm pleased to inform you that your manuscript has been deemed suitable for publication in PLOS ONE. Congratulations! Your manuscript is now with our production department. 

Kind regards, 

on behalf of

Dr. Rajakumar Anbazhagan 

Academic Editor

PLOS ONE